# A novel pathway of LPS uptake through syndecan-1 leading to pyroptotic cell death

Shigetoshi Yokoyama[1], Yan Cai[1], Miyuki Murata[1], Takeshi Tomita[1], Mitsuhiro Yoneda[1], Lei Xu[1], Aprile L Pilon[2], Raul E Cachau[3], Shioko Kimura[1]*

[1]Laboratory of Metabolism, National Cancer Institute, National Institutes of Health, Bethesda, United States; [2]APCBIo Innovations Inc., Rockville, United States; [3]Advanced Biomedical Computing Center, Frederick National Laboratory for Cancer Research, Leidos Biomedical Inc., Frederick, United States

**Abstract** Intracellular lipopolysaccharide (LPS) triggers the non-canonical inflammasome pathway, resulting in pyroptosis of innate immune cells. In addition to its well-known proinflammatory role, LPS can directly cause regression of some tumors, although the underlying mechanism has remained unknown. Here we show that secretoglobin(SCGB)3A2, a small protein predominantly secreted in airways, chaperones LPS to the cytosol through the cell surface receptor syndecan-1; this leads to pyroptotic cell death driven by caspase-11. SCGB3A2 and LPS co-treatment significantly induced pyroptosis of macrophage RAW264.7 cells and decreased cancer cell proliferation in vitro, while SCGB3A2 treatment resulted in reduced progression of xenograft tumors in mice. These data suggest a conserved function for SCGB3A2 in the innate immune system and cancer cells. These findings demonstrate a critical role for SCGB3A2 as an LPS delivery vehicle; they reveal one mechanism whereby LPS enters innate immune cells leading to pyroptosis, and they clarify the direct effect of LPS on cancer cells.
DOI: https://doi.org/10.7554/eLife.37854.001

*For correspondence:
kimuras@mail.nih.gov

## Introduction

The airway is continuously exposed to pathogens, including low levels of gram negative bacteria in the air (*Lundin and Checkoway, 2009*). Lipopolysaccharide (LPS) is a component of the outer membrane of gram negative bacteria and can cause inflammation in the lung. It was previously thought that toll-like receptor 4 (TLR4) is the sole LPS-specific pattern recognition receptors (PRRs) at the cell membrane (*Poltorak et al., 1998*). However, recent studies demonstrated the presence of an TLR4-independent PRRs mechanism to sense LPS in the cytosol via an inflammatory caspase, caspase-11, in a non-canonical inflammasome pathway (*Hagar et al., 2013*; *Kayagaki et al., 2013*). While it is widely known that tumor metastasis is coupled with inflammation in the tumor microenvironment, in many cases, immune cells in the tumor microenvironment no longer exhibit anti-tumor effects, instead they are co-opted to promote tumor growth and metastasis (*Whiteside, 2008*). On the contrary, the activity of bacteria or endotoxin for anti-tumor effects has been extensively studied for decades since the first observation by W. B. Coley (*Lundin and Checkoway, 2009*; *Ribi et al., 1983*). Although 'Coley's toxin' is currently not used for cancer treatment because of its toxicities, accumulating evidence has revealed that his theory was correct and the notion that the enhanced host immune systems by endotoxin could attack some cancer cells has advanced to cancer immunotherapy. However, whether endotoxin has a direct function in attacking cancer cells remains controversial, while the interest in endotoxin as a cancer therapeutic agent waned, despite of many reports for favorable outcomes.

**eLife digest** Inflammation serves to kill invading bacteria and viruses. Certain molecules on the surface of the microbes can trigger an inflammatory cascade, and one example of such a molecule is lipopolysaccharide (LPS). Cells can react to LPS by triggering a process called pyroptosis that causes the cell to burst and die. The released cell contents attract blood and lymphatic cells that in turn kill the LPS-producing bacteria. This prevents the bacteria from multiplying and spreading.

LPS was used in the very early days of medicine to treat cancer, although it has fallen out of favor because it causes severe side effects, such as a hyperinflammatory response (sepsis) that can result in death. It was not known exactly how LPS kills cancer cells, which has limited its use. Yokoyama et al. now show that a protein called SCGB3A2, which is produced by the cells that line the lung airways, binds to LPS. Tests on mouse immune cells and lung cancer cells grown in the laboratory showed that the resulting SCGB3A2-LPS complex can then bind to a cell surface protein called syndecan 1. This enables LPS to enter the cell and trigger pyroptosis and cell death.

To confirm the role of SCGB3A2 in pyroptosis, Yokoyama et al. examined tumor growth in mice that are not able to produce SCGB3A2. These mice developed more tumors than normal mice, but tumor growth was suppressed when mice were injected with SCGB3A2.

The findings presented by Yokoyama et al. could potentially lead to new types of cancer treatments, particularly for lung cancers. However, it remains to be examined whether molecules that trigger pyroptosis, like LPS, could also be used to treat cancers other than those from the lung. Further work is also needed to understand in more detail how SCGB3A2 and LPS work together to cause cancer cell death.

DOI: https://doi.org/10.7554/eLife.37854.002

A cytokine-like small secreted protein, SCGB3A2 (secretoglobin 3A2, also known as UGRP1 and HIN-2), was previously identified that is abundantly and specifically expressed in non-ciliated airway epithelial (club) cells of the trachea, bronchus, and bronchioles (*Niimi et al., 2001*) and revealing that SCGB3A2 functions to suppress lung inflammation and fibrosis (*Cai and Kimura, 2015*; *Cai et al., 2014*; *Chiba et al., 2006*; *Kido et al., 2014*; *Kurotani et al., 2011*; *Yoneda et al., 2016*). Although specific expression of SCGB3A2 in lung epithelial cells and its role in inflammation may imply a possible important function for SCGB3A2 in the clearance of pathogens, its role in host defense, if any, has not been studied. In addition, while fibrosis is closely related to tumor development (*Coussens and Werb, 2002*; *Trinchieri, 2012*) and SCGB3A2 functions as an anti-fibrotic agent, the role of SCGB3A2 in lung cancer development is unknown.

## Results

### SCGB3A2 inhibits LLC cell growth in vitro and in vivo

To determine whether SCGB3A2 has any influence on cancer cell growth, CCK8 (cell counting kit 8) assay was performed using murine Lewis lung carcinoma (LLC) cells. The proliferation of LLC cells was markedly suppressed by mouse recombinant SCGB3A2 (*Figure 1A*). This in vitro effect of SCGB3A2 was also observed in vivo in the LLC cells intravenous metastasis model using wild-type C57BL/6 mice in conjunction with administration of SCGB3A2 (*Figure 1B–1E*). To confirm the tumor growth inhibition roles of SCGB3A2 in vivo, *Scgb3a2*-null mice were subjected to the metastasis model (*Kido et al., 2014*). Mice null for *Scgb3a2* developed far greater numbers of lung surface tumors than wild-type littermates when LLC cells were intravenously injected (*Figure 1F*). Furthermore, administration of recombinant mouse SCGB3A2 to *Scgb3a2*-null mice clearly rescued the *Scgb3a2*-null phenotypes of LLC cell lung metastasis (*Figure 1G–1I*). These results indicate the importance of SCGB3A2 in the suppression of LLC cell tumor development in lungs in vivo.

### SCGB3A2 binds to LPS

For the above experiments, several preparations of recombinant SCGB3A2 (mouse and human) were used from various sources as described in Materials and methods. However, we noticed an unexpected phenomenon where some sources of recombinant SCGB3A2 had almost no effect on

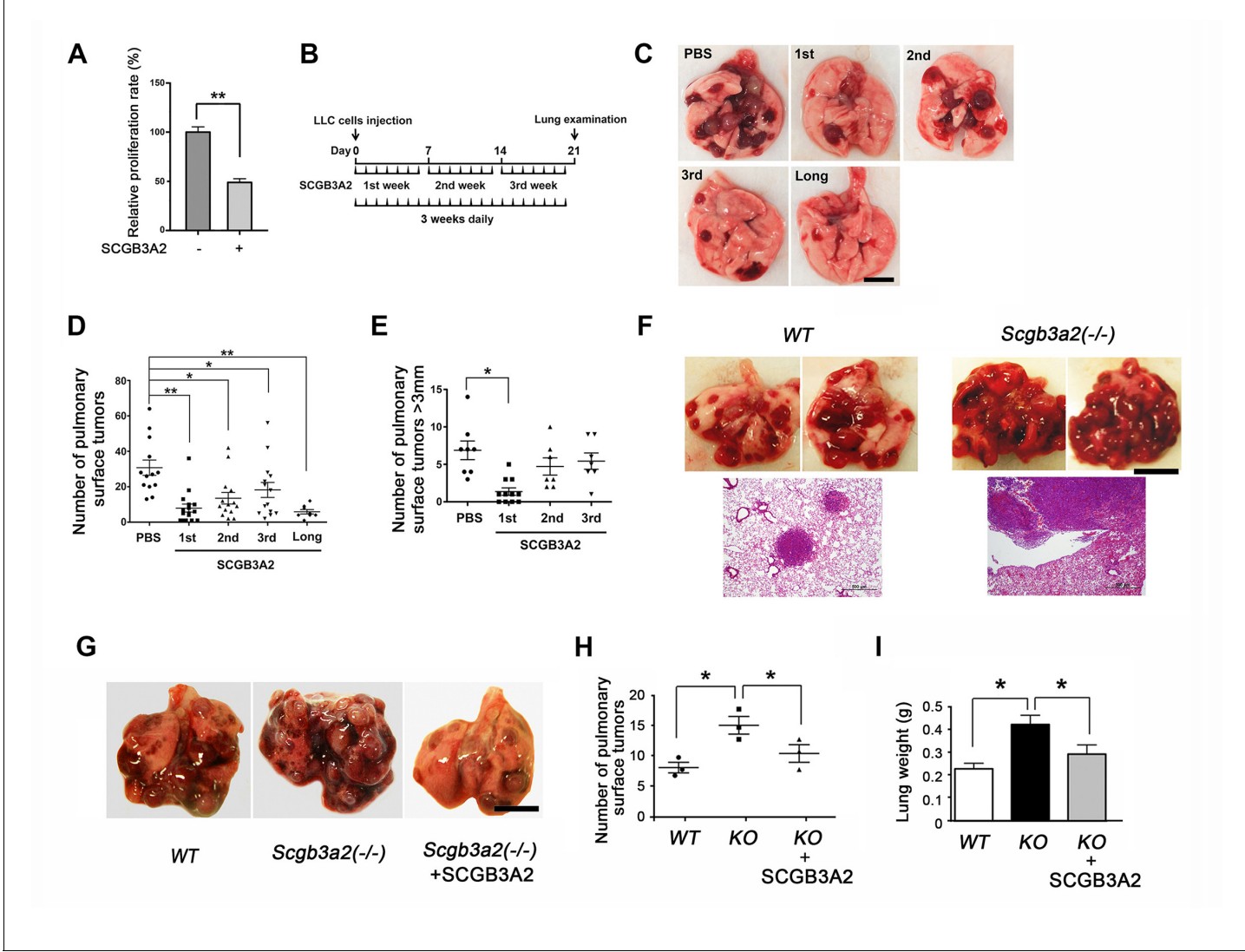

**Figure 1.** SCGB3A2-induced suppression of LLC cell proliferation. (**A**) Effect of SCGB3A2 on proliferation of LLC cells. Cells were maintained without serum for 24 hr, followed by 1% FBS-RPMI1640 media with or without mouse SCGB3A2 (1 µg/ml). CCK8 assay was carried out 72 hr after the addition of SCGB3A2. Averages ± SD from three independent experiments, each in triplicate. **p<0.01 by student's *t*-test. (**B**) LLC cell intravenous metastasis model scheme. Mice inoculated with LLC cells received daily intravenous administration of mouse recombinant SCGB3A2 for 7 consecutive days for the 1st, 2nd, or 3rd week, or the entire experimental period of 21 days (long). Control mice received PBS alone. The number of the pulmonary surface tumors was counted on day 21. N = 7–14 per group. (**C**) Representative lung images from each SCGB3A2 administration group. Scale bar = 1 cm. (**D**) Summary for the numbers of pulmonary surface tumors. A dot indicates a mouse. Averages ± SD are shown. *p<0.05, **p<0.01. (**E**) Number of pulmonary surface tumors larger than 3 mm. Averages ± SD are shown. *p<0.05. (**F**) (upper panel) Representative pictures of metastasized lung tumors from wild-type (WT) and *Scgb3a2*-null (*Scgb3a2*(-/-)) mice. Scale bar = 1 cm. (lower panel) H & E staining of lungs. Scale bar = 500 µm. (**G**) Representative lungs from wild-type littermate (WT), *Scgb3a2*-null (Scgb3a2(-/-)), and *Scgb3a2*-null mice given SCGB3A2 (ie. Scgb3a2(-/-)+SCGB3A2) for the 1st week. N = 3 per group. This was a separate independent experiment from those presented in C-E. WT and *Scgb3a2*(-/-) mice received daily PBS as a control. Lung necropsy was carried out on day 21. (**H**) Graph showing the number of pulmonary surface tumors of experiment in G. Averages ± SD are shown. (**I**) Lung weights of each LLC cell metastasis model in G. Averages ± SD are shown. KO: *Scgb3a2*-null. *p<0.05. Statistical differences calculated by One-way ANOVA except in A.

DOI: https://doi.org/10.7554/eLife.37854.003

LLC cell growth inhibition (*Figure 2—figure supplement 1A*). This phenomenon was found to be associated with the level of endotoxin (LPS) contained in the various preparations (*Supplementary file 1*). Moreover, we realized that whenever the endotoxin was removed from the preparation, the final SCGB3A2 protein yield was drastically reduced (data not shown). Because

SCGB3A2 is abundantly expressed in airway epithelial cells which are exposed to various microorganisms and LPS derived from bacteria (*Lundin and Checkoway, 2009*), we hypothesized that the fundamental function of SCGB3A2 may be related to its binding to and sequestering of LPS. Further, while inflammation is thought to be coupled with cancer metastasis, paradoxically endotoxins or ensuing enhanced immunity may inhibit some cancer growth (*Lundin and Checkoway, 2009*; *McCarthy, 2006*; *Ribi et al., 1983*). Indeed, the growth of LLC cells was strongly inhibited by small amounts of LPS (*Figure 2A*). To test if SCGB3A2 interacts with LPS, imidazole and zinc salt staining was performed (*Figure 2B and C*) (*Rodríguez and Hardy, 2015*). Crude LPS (O111:B4) barely migrated into the gel and remained near the well, due to high-molecular mass aggregation (*Figure 2B*, lane 1 (*Rodríguez and Hardy, 2015*)). Pre-incubation with SCGB3A2 (human SCGB3A2 (C1); see *Supplementary file 1*; unless otherwise noted, this lot was mainly used in the current studies) produced a broad diffuse band in dose dependent manner, indicating that SCGB3A2 interacted with and dramatically enhanced the electrophoretic mobility of LPS (*Figure 2B*, lane 2–5, and *Figure 2C*). Rough A form (Ra-LPS) and other serotypes of LPS produced the same results (*Figure 2—figure supplement 1B*). To further confirm that SCGB3A2 directly binds to LPS, streptavidin pull-down assays were performed using LPS-Biotin and recombinant SCGB3A2 (*Figure 2D*). The results clearly showed that SCGB3A2 is an LPS binding protein. The ability of SCGB3A2 to bind and disaggregate LPS micelles was further demonstrated by the dynamic light scattering (DLS) method (*Figure 2E* and *Figure 2—figure supplement 1C–1E*). Thus, SCGB3A2 is an LPS binding protein and has powerful LPS dissociation properties, against both smooth and rough forms of LPS.

To determine if LPS alone is sufficient or the combination of LPS+SCGB3A2 is required for the inhibition of growth and metastasis of LLC cells in vivo, LLC cell intravenous metastasis xenograft experiments were carried out, in which various amounts of LPS, estimated in our recombinant protein SCGB3A2 preparations (see *Supplementary file 1*), were administered for seven consecutive days in the 1 st week after LLC cells injection (see *Figure 1B*). The number of tumors obtained was compared with that obtained with administration of recombinant SCGB3A2 without exogenously added LPS. Human SCGB3A2(C1) alone showed drastic inhibition of LLC cells growth, while the amount of LPS contained in the recombinant SCGB3A2 preparation C1 or C3, or high concentration did not show any statistically significant differences in tumor numbers compared with PBS administration (*Figure 2F*). Moreover, LPS-treated lungs showed much larger lesions than did SCGB3A2-treated lung tumors, which sometimes encompassed the entire lobes, demonstrating the fundamental differences between LPS alone and SCGB3A2 administration (*Figure 2G*).

## SDC1 is a receptor for SCGB3A2

A receptor for SCGB3A2 involved in the SCGB3A2 signaling was unbiasedly identified using human protein microarray analysis (*Figure 3A*, *Supplementary file 2* and *3*, *Figure 3—source data 1*). Among the top 116 proteins (*Supplementary file 2*), 13 proteins were selected as possible candidates for the SCGB3A2 receptor as a cell surface protein (*Figure 3A* and *Supplementary file 3*). To confirm a direct interaction with SCGB3A2, pull-down assays were performed, in which syndecan-1 (SDC1) and podoplanin (PDPN, T1-alpha) showed positive interaction with SCGB3A2 (*Figure 3B* and data not shown). PDPN is known as a marker for alveolar type I epithelial cells in lung, while SDC1 was moderately expressed in proximal airway epithelial cells (*Figure 3C*), suggesting a possible relationship between SCGB3A2 and SDC1 for lung airway homeostasis. Therefore, this study focused on SDC1.

SDC1 was found to be highly expressed on LLC cells surfaces in vitro as well as in metastatic LLC cells in vivo (*Figure 3D* and *Figure 3—figure supplement 1A*). In contrast, the B16F10 mouse melanoma cell line, which exhibited less SCGB3A2-dependent growth suppression effects than LLC in vitro (data not shown), showed focal expression of SDC1 near cell nuclei and faint staining at cell-to-cell contact sites (*Figure 3D*), while the total cell surface staining was low compared to LLC cells. Further analyses supported the robust expression of SDC1 on the surface of LLC cells (*Figure 3E* and *Figure 3—figure supplement 1B*), and their binding to SCGB3A2 (*Figure 3F*). LLC cells stably expressing shRNA-SDC1 (LLC-sh-SDC1, *Figure 3—figure supplement 1C*) showed diminished SCGB3A2 binding (*Figure 3G*). In addition, ARH-77 human myeloma cell line, which lacks detectable SDC1 (*Ridley et al., 1993*), and ARH-77 cells over-expressing mouse SDC1 (ARH-77-mSDC1 (*Dhodapkar et al., 1998*; *Liebersbach and Sanderson, 1994*), See *Figure 3—figure supplement 1D*) verified the SCGB3A2-SDC1 binding interaction. ARH-77-mSDC1 enhanced SCGB3A2 binding

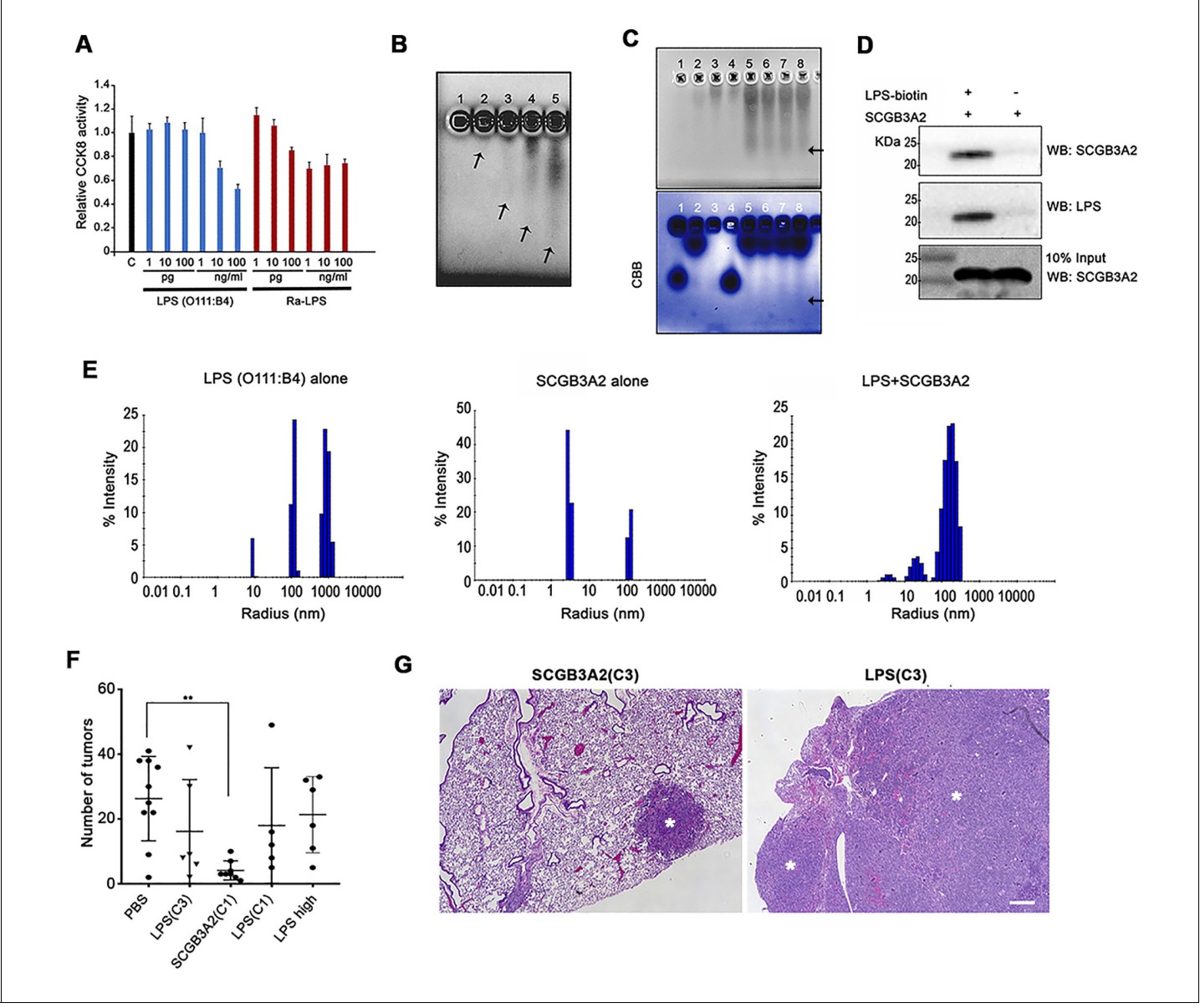

**Figure 2.** SCGB3A2 as an LPS binding protein. (**A**) CCK8 analysis using various concentrations as shown in the bottom (pg, ng/ml) of smooth LPS (*E.coli* O111:B4 serotype) and rough LPS (Ra-LPS) after 72 hr in culture. C; control without any addition of LPS. Averages ± SD from three independent experiments, each in triplicate. (**B**) Reverse staining of aggregation of LPS. Imidazole-zinc staining of *E.coli* O111:B4 serotype LPS on agarose gel. LPS (10 µg) was incubated with human SCGB3A2 in lane 1 to 5: 0, 10, 100 ng, 1, and 10 µg, respectively. Arrows indicate the bottom of the aggregate or smeary bands. (**C**) Reverse staining of aggregation of LPS. Imidazole-zinc staining of *E.coli* O111:B4 serotype LPS on agarose gel. BSA 10 µg (lane1), human SCGB3A2 10 µg (lane 2), LPS 10 µg (lane 3), BSA +LPS pre-incubation at 37 °C, 30 min (lane 4), SCGB3A2 + LPS pre-incubation at 37 °C, 30 min (lane 5), SCGB3A2 + LPS pre-incubation at RT, 30 min (lane 6), SCGB3A2 + LPS pre-incubation at 37 °C, 10 min (lane 7), SCGB3A2 + LPS pre-incubation at RT, 10 min (lane 8). Bottom image is Coomassie Brilliant Blue (CBB) staining of the same gel. Arrows indicate the bottom of the aggregate or smeary bands. (**D**) Streptavidin pull-down assay of LPS-Biotin and recombinant SCGB3A2. IP and western blotting were sequentially carried out using anti-SCGB3A2 and anti-LPS antibody, respectively. Input is 10%. (**E**) DLS assay. Size deformation of LPS micelles by human SCGB3A2 pre-incubation. Histogram shows the intensity of hydrodynamic radii (nm) of O111:B4 LPS (20 µg/ml), human SCGB3A2 (20 µg/ml), and LPS pre-incubated with SCGB3A2 for 30 min at RT. Gel analysis and DLS assay were carried out more than 3 separate times and each time, similar results were obtained. (**F**) Effect of SCGB3A2 or LPS on the number of lung surface tumors in LLC cell intravenous metastasis model. LPS(C3): LPS concentration equivalent to that contained in mouse SCGB3A2(C3) (see *Figure 1* and *Supplementary file 1*), SCGB3A2(C1): human SCGB3A2(C1) protein without addition of exogenous LPS, LPS(C1): LPS concentration equivalent to contained in human SCGB3A2(C1), and LPS high: LPS (1 µg/mouse). A dot indicates a mouse. Averages ± SD are shown. **p<0.01. (**G**) Representative images of lung of mice with SCGB3A2(C3) or LPS(C3) administration. Asterisks indicate tumors. Bar = 300 µm.

*Figure 2 continued on next page*

*Figure 2 continued*

DOI: https://doi.org/10.7554/eLife.37854.004

The following figure supplement is available for figure 2:

**Figure supplement 1.** Analysis of LPS-SCGB3A2 complex.

DOI: https://doi.org/10.7554/eLife.37854.005

compared to the parental cells (*Figure 3—figure supplement 1E*). Syndecans are a family of transmembrane heparan sulfate proteoglycan (HSPG). To determine the domain of SDC1 that interacts with SCGB3A2, heparin was used to inhibit the function of heparan sulfate chains (HS). Heparin addition significantly inhibited the binding of SCGB3A2 to both LLC (*Figure 3H*) and ARH-77-mSDC1 cells (*Figure 3—figure supplement 1F*), suggesting that HS on SDC1 may play a role in SCGB3A2 binding.

## SCGB3A2 accumulates on the uropod and is incorporated through clathrin-mediated endocytosis in LLC cells

To reveal the precise binding site of SCGB3A2 on LLC cell surfaces, immunofluorescence analysis was performed using anti-SCGB3A2 and anti-SDC1 ectodomain antibodies (*Figure 4A*). ARH-77-mSDC1 cells were also used in this analysis. Without stimulation with SCGB3A2, both LLC and ARH-77-mSDC1 cells had evenly distributed SDC1 on the cell surface (*Figure 4A* Control). After stimulation with SCGB3A2, the SDC1 signal became relatively concentrated on cell protrusions equivalent to the uropod structure of myeloma (*Børset et al., 2000*; *Yang et al., 2003*), which co-localized with SCGB3A2 (*Figure 4A*,+SCGB3A2). Staining of ICAM-1, a uropod marker (*del Pozo et al., 1997*), confirmed co-localization of SDC1 and SCGB3A2 on the uropods of both LLC and ARH-77-mSDC1 cells. Interestingly, when LLC cells were incubated for a short time with LPS and SCGB3A2, Alexa-labeled LPS (LPS$_{A488}$,*Figure 4—figure supplement 1A*), SCGB3A2, and clathrin, a key protein for endocytosis, all co-localized in uropod (*Figure 4B*). This pattern of clathrin localization was similar to those previously reported using T lymphocyte (*Samaniego et al., 2007*). Upon further incubation, LLC cells appeared to have incorporated SCGB3A2 into the cells as visualized using an HaloTag (HT)-SCGB3A2 fusion protein (*Figure 4—figure supplement 1B and C*). Clathrin expression was localized near the incorporated SCGB3A2 signals (*Figure 4—figure supplement 1C*), suggesting that the LPS-SCGB3A2 complex enters LLC cells via binding to SDC1 followed by clathrin-dependent endocytosis (see below). Further, live cell imaging clearly showed that SCGB3A2-HT was incorporated into LLC-sh-Control cells after overnight incubation, while very low signals were observed in LLC-sh-SDC1 cells (*Figure 4—figure supplement 1D*). Computer modeling analysis provided further evidence that SCGB3A2 binds to both LPS and SDC1 when it forms a tetramer (*Figure 4C* and *Figure 4—figure supplement 2A–2F*). In fact, SCGB3A2 tends to form oligomers in vitro, demonstrating the validity of the computer modeling (*Figure 4—figure supplement 2G*, see *Figure 2C* CBB staining, also cf: (*Cai et al., 2014*; *Niimi et al., 2001*)).

## SCGB3A2 functions as a chaperone to deliver LPS into the cytosol and activates caspase-11/NLRP3 inflammasome foci formation

Recent studies demonstrated intracellular LPS triggers caspase-4/11 activation and the non-canonical inflammasome pathway (*Hagar et al., 2013*; *Kayagaki et al., 2013*). It's possible that SCGB3A2 simply enhances TLR4 priming canonical signals via SDC1 binding, transferring LPS to TLR4. To address this possibility, LLC cells stably expressing sh-TLR4 (LLC-sh-TLR4) were established (*Figure 4—figure supplement 3A*), and SCGB3A2 binding and uptake were compared with those of LLC-sh-Control and LLC-sh-SDC1 cells (*Figure 4—figure supplement 3B and C*). SCGB3A2 enhanced binding of LPS to LLC-sh-TLR4 cells at similar level to that of LLC-sh-Control, while LLC-sh-SDC1 cells showed little binding of LPS (*Figure 4—figure supplement 3B*). In addition, SCGB3A2 and LPS were incorporated into LLC-sh-TLR4 cells and appeared to co-localize within the cytosol (*Figure 4—figure supplement 3C*). These data, together with data in *Figure 4—figure supplement 1D*, suggest that SCGB3A2 is important for LPS uptake and that SDC1, not TLR4, is required for SCGB3A2-LPS incorporation.

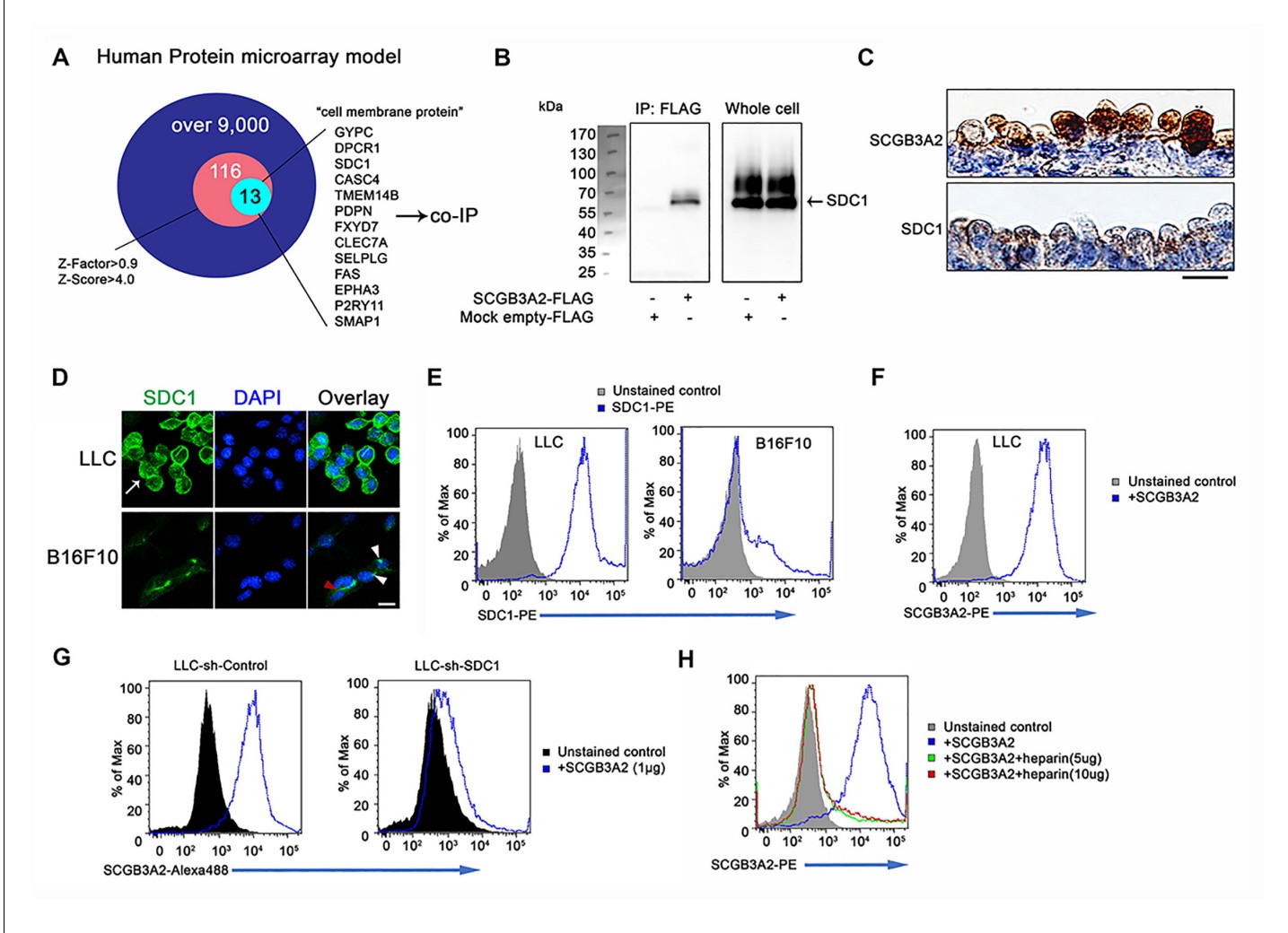

**Figure 3.** SCGB3A2 binding to LLC cells through HS of SDC1. (A) Schematic model of a human protein array for the isolation process of candidate genes shown as a Venn diagram. (B) Co-immunoprecipitation assay of SCGB3A2-FLAG and SDC1-Myc-His in COS-1 cells. IP and western blotting were sequentially carried out using anti-FLAG and anti-Myc antibody, respectively. (C) SCGB3A2 and SDC1 immunostaining in the airway epithelial cells of adult wild-type mouse lungs. Counterstained with Hematoxylin. Bar = 10 μm. (D) Immunofluorescent staining of SDC1 in LLC and B16F10 cells grown in 10%FBS-RPMI 1640 medium for 24 hr. DAPI was used for nuclear staining. White arrow indicates SDC1 cell surface expression in LLC. Red arrowhead points to the staining at cell-cell junction and white arrowheads point to focal SDC1 staining near the nucleus in B16F10 cells. Bar = 20 μm. (E) Flow cytometric analysis for SDC1 expression on cell surfaces of LLC and B16F10 cells using PE conjugated anti-SDC1 ectodomain specific antibody. (F) Flow cytometric analysis for SCGB3A2 binding to LLC cells using anti-SCGB3A2 antibody. GST tagged mouse SCGB3A2 (3 μg) was incubated with LLC cells at 4 °C for 30 min. Cells were stained with rabbit anti-mouse SCGB3A2 antibody, followed by staining with PE-anti-rabbit IgG antibody at 4 °C for 30 min. (G) SCGB3A2 binding assay on LLC-sh-Control or sh-SDC1 cells. GST tagged mouse SCGB3A2 (1 μg) was incubated with each cell type at 4 °C for 30 min, followed by staining with Alexa 488 anti-rabbit IgG antibody at 4 °C for 30 min. (H) SCGB3A2 binding assay on LLC cells. Cells were co-incubated with or without GST tagged mouse SCGB3A2 (1 μg) or SCGB3A2 + heparin. Cells were stained with PE-anti-rabbit IgG antibody at 4 °C for 30 min. Data except A are the representative from more than three independent experiments.

DOI: https://doi.org/10.7554/eLife.37854.006

The following source data and figure supplement are available for figure 3:

**Source data 1.** Results of whole protein-protein interaction array.

DOI: https://doi.org/10.7554/eLife.37854.008

**Figure supplement 1.** Cell surface expression of SDC1 and validation of sh-SDC1 and ARH-77-mSDC1 clones.

DOI: https://doi.org/10.7554/eLife.37854.007

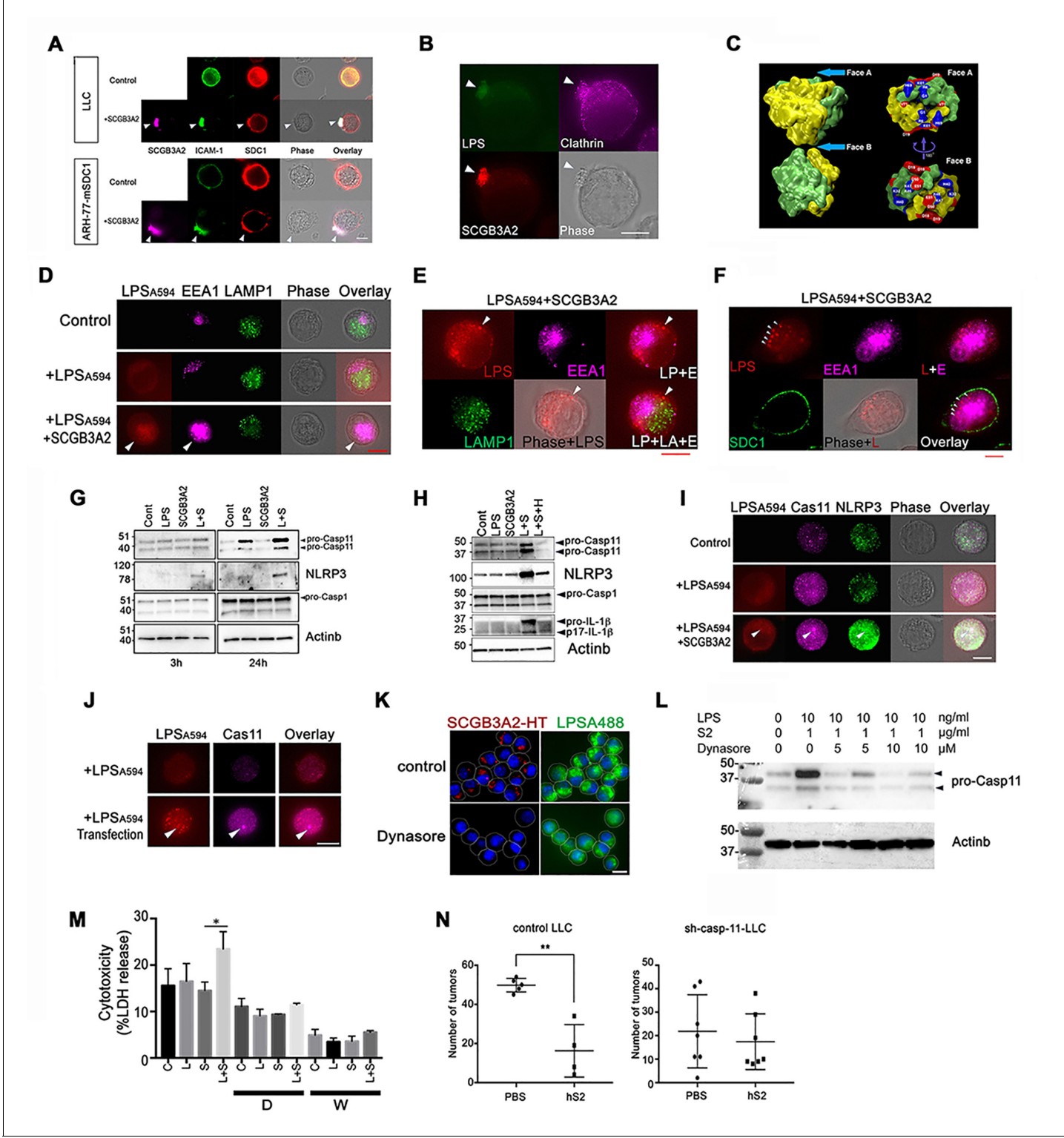

**Figure 4.** SCGB3A2-LPS uptake activates inflammasome signaling. (**A**) Immunofluorescence analysis of LLC and ARH-77-mSDC1 cells for SCGB3A2, SDC1, and ICAM-1. Arrowheads: uropod-like structures. Cells were incubated in 0% FBS-RPMI 1640 for 40 min (0%40 m) with or without GST-mSCGB3A2 (1 μg). Scale bar = 10 μm. (**B**) Immunofluorescence analysis of LLC cells for clathrin and HT (HaloTag). Cells were incubated with LPS$_{A488}$ (1 μg/ml) and SCGB3A2-HT supernatant (SN) in 0%1 hr. Arrowheads: uropod structure. Bar = 10 μm. (**C**) SCGB3A2 tetramer model (see *Figure 4—figure supplement 2A–2F*). Left: Exploded view of the tetramer model showing the two dimers, which are colored to identify the two monomers. The dimer structure reveals a pocket accessible from Face A and flanked by positively charged residues (Gly 1 N termini-; Arg 6 and Lys 61) forming a pattern

*Figure 4 continued on next page*

*Figure 4 continued*

consistent with a heparan or LPS binding motif. Face A is exposed in the tetramer while Face B (right) is occluded. (**D**) Immunofluorescence analysis of LLC cells maintained in 0%5 hr for EEA1, LAMP1 and LPS$_{A594}$. Arrowhead: overlapping staining of LPS$_{A594}$ and EEA1. Bar = 10 μm. (**E**) Immunofluorescence analysis of LLC cells after 1%16 hr incubation with LPS$_{A594}$ (2 μg/ml) and SCGB3A2 (2 μg/ml). Arrowheads: cytosolic LPS signal, not overlapping with EEA1 and LAMP1 staining. LP; LPS, E; EEA1, LA; LAMP1. Bar = 10 μm. (**F**) Immunofluorescence analysis of LLC cells for LPS$_{A594}$, EEA1, SDC1 after 1%16 hr incubation with LPS$_{A594}$ (2 μg/ml) and SCGB3A2 (2 μg/ml). Arrowheads: cytosolic LPS signal, not overlapping with EEA1 and SDC1 staining. Bar = 10 μm. (**G**) Immunoblots of LLC-sh-TLR4 cells after treatment with or without LPS (O111:B4, 10 ng/ml) or SCGB3A2 (200 ng/ml) for 3 and 24 hr in OPTI-MEM. (**H**) Immunoblots of RAW264.7 cells after treatment with or without LPS (O111:B4, 1 μg/ml), SCGB3A2 (1 μg/ml), and/or heparin (1 μg/ml) for 16 hr in OPTI-MEM. Abbreviations for G and H, L: LPS, S: SCGB3A2, H: Heparin. Immunoblots shown in G and H were carried out more than 3 separate times, and each time similar results were obtained. (**I**) Immunofluorescence analysis of LLC cells for caspase-11 and NLRP3, incubated for 0 %5 hr with or without LPS$_{A594}$ (2 μg/ml), LPS$_{A594}$ +SCGB3A2 (2 μg/ml). Arrowheads: LPS, caspase-11, NLRP3 overlapping focus. Bar = 10 μm. (**J**) Immunofluorescence analysis of caspase-11 in LPS transfected LLC. LPS$_{A594}$ (1 μg) was transfected using X-tremeGENE HP in 10%5 hr. Arrowheads: LPS and caspase-11 focus. Bar = 10 μm. All images are the representative of three independent experiments. (**K**) Immunofluorescence analysis of LLC cells for SCGB3A2-HT and LPS$_{A488}$. Cells were incubated for 1%24 hr with or without Dynasore (10 μM), and LPS$_{A488}$ (1 μg)+SCGB3A2-HT supernatant. Dotted lines depict the outer membrane of cells. Bar = 10 μm. (**L**) Immunoblots of LLC cells after treatment with or without LPS (O111:B4, 10 ng/ml), SCGB3A2 (1 μg/ml) or Dynasore (5 or 10 μM) for 2%3 hr. (**M**) LDH release from LLC cells in control condition or in the presence of Dynasore (D, 5 μM) or Wedelolactone (W, 10 μM), with or without LPS (1 μg/ml) and/or SCGB3A2 (1 μg/ml) for 2%48 hr. Average ±SD from three independent experiments, each in triplicate. S: SCGB3A2, L + S; LPS +SCGB3A2. *: p<0.05 by one-way ANOVA. (**N**) Summary for the numbers of pulmonary surface tumors in lungs of intravenous metastasis model mice using LLC-control or LLC-sh-Casp-11 cells and indicated treatments. hS2; human SCGB3A2. A dot indicates a mouse. **: p<0.01 by student's *t*-test.

DOI: https://doi.org/10.7554/eLife.37854.009

The following figure supplements are available for figure 4:

**Figure supplement 1.** Analysis of SCGB3A2-HaloTag protein trafficking.

DOI: https://doi.org/10.7554/eLife.37854.010

**Figure supplement 2.** SCGB3A2 modeling.

DOI: https://doi.org/10.7554/eLife.37854.011

**Figure supplement 3.** Establishment of LLC-sh-TLR4 cells and analysis of SCGB3A2/LPS binding/incorporation.

DOI: https://doi.org/10.7554/eLife.37854.012

**Figure supplement 4.** Effect of SCGB3A2-LPS on RAW264.7 cells.

DOI: https://doi.org/10.7554/eLife.37854.013

**Figure supplement 5.** Confirmation of caspase-11 knockdown in LLC cells in the presence of Wedelolactone or LLC-sh-casp11 cells.

DOI: https://doi.org/10.7554/eLife.37854.014

To confirm the cytosolic localization of LPS, LLC cells were visualized with Alexa-labeled LPS (LPS$_{A594}$), anti-EEA1 (early endosomes marker) and anti-Lamp1 (lysosomal marker) antibodies (*Figure 4D and E*). At an early time point, most of the LPS staining was co-localized with EEA1, which was clearly enhanced when cells were co-incubated with SCGB3A2 (*Figure 4D*). At a later time point, however, some of the LPS staining did not overlap with either EEA1 or LAMP1 (*Figure 4E*). With SDC1 staining depicting the plasma membrane, LPS staining was clearly visible inside the membrane, which differed from the EEA1 distribution pattern (*Figure 4F*). These results suggest that LPS could be transported into the cytosol of LLC cells through an SCGB3A2-dependent mechanism.

Next whether LPS transport into cytosol of LLC cells triggers non-canonical inflammasome pathway was examined using LLC-sh-TLR4 cells. This was because TLR4 signaling also enhances pro-caspase-11/NLRP3 expression via the canonical inflammasome pathway (*Kayagaki et al., 2013*). SCGB3A2 + LPS increased pro-caspase-11 and NLRP3, while caspase-1 expression, the major caspase activated by the canonical inflammasome, was not significantly different (*Figure 4G*). Caspase-11 processing and IL-1β expression/processing were not detected at the protein level in LLC-sh-TLR4 and LLC-sh-Control cells (data not shown). This might not exclude the possibility that SCGB3A2 promotes pyroptosis of LLC cells with only small amounts of the processed form of caspase-11 that cannot be detected by western blotting based on previous reports (*Hagar et al., 2013*) (see *Figure 5A*).

The importance of sensing LPS and triggering caspase-11 and NLRP3 activation in host defense has been mainly studied using macrophage. Moreover, macrophage are key players both for lung homeostasis and the tumor microenvironment. Hence, the effect of SCGB3A2 on mouse macrophage-like RAW264.7 cells, which express SDC1 on the cell surface (*Figure 4—figure supplement*

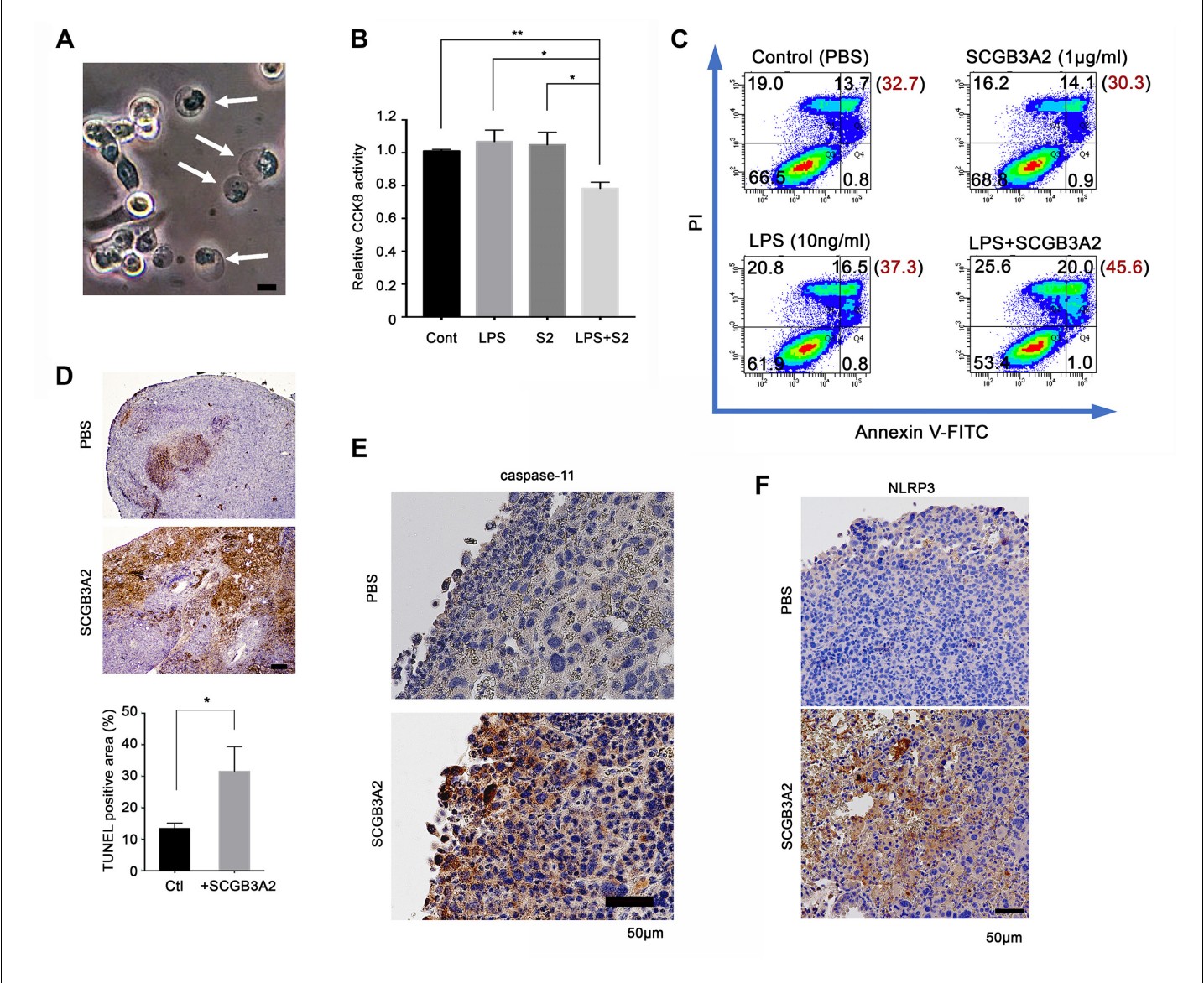

**Figure 5.** SCGB3A2-LPS promotion of pyroptotic cell death. (**A**) Representative phase contrast image of LLC cells incubated with LPS (O111:B4, 1 μg/ml) and SCGB3A2 (1 μg/ml) in 1% FBS-RPMI 1640 for 72 hr. Arrows indicate cells undergoing pyroptosis (swollen cells). Bar = 10 μm. Image is the representative of three independent experiments (**B**) CCK8 analysis using LLC cells with LPS (O111:B4) (1 ng/ml) and/or human SCGB3A2 (C2; see *Supplementary file 1*) (10 ng/ml) in 2%FBS-RPMI for 72 hr culture. Data are the representative from more than three independent experiments, each in triplicate. Averages ± SD are shown. (**p<0.01, *p<0.05 by One-way ANOVA). S2; SCGB3A2. (**C**) Flow cytometry analysis for Annexin V/PI staining. 1 μg/ml SCGB3A2 and 10 ng/ml LPS (O111:B4) were used. LLC cells were maintained for 24 hr in 1%FBS-RPMI medium, then treated with or without SCGB3A2/LPS, and further cultured for 48 hr. The numbers in the graph indicate the cell percentage in each quadrants. The number in blanket show the total percentage of PI positive cells in each graph. Experiments were carried out more than 3 times, and each time similar results were obtained. (**D**) TUNEL staining of lung sections of lung metastasized LLC cells in the intravenous administration model. Images are shown for control (PBS) and mouse SCGB3A2 administered during the 2nd week. The bottom graph indicates the percentage of TUNEL positive areas per total LLC tumor areas as measured using imageJ. +SCGB3A2 indicates lungs of mice that received SCGB3A2 during the 1 st week. Ctl, control. Three independent lung samples were evaluated for each group. *p<0.05 by student's t-test. (**E**) Representative IHC staining of caspase-11 for metastasized nodules of LLC cells in lung in vivo. Images are shown for control (PBS) and mouse SCGB3A2 administered during the 1 st week. Counterstained with Haematoxylin. Bar = 50 μm. (**F**) Representative IHC staining of NLRP3 for metastasized nodules of LLC cells in lung. Images are shown for control (PBS) and mouse SCGB3A2 administered during the 1 st week. Counterstained with Haematoxylin. Bar = 50 μm. IHC staining was carried out using more than three independent lung samples.

DOI: https://doi.org/10.7554/eLife.37854.015

*4A*), was next examined. Co-incubation of RAW264.7 cells with SCGB3A2 + LPS clearly enhanced expression of caspase-11 and NLRP3, followed by IL-1β up-regulation and maturation, while SCGB3A2 or LPS alone did not (*Figure 4H*). Heparin co-incubation abrogated caspase-11/NLRP3/IL-1β expression by SCGB3A2 + LPS. SCGB3A2 enhanced IL-1β secretion from RAW264.7 cells, which was inhibited by the addition of heparin (*Figure 4—figure supplement 4B*). A lactate dehydrogenase (LDH) cytotoxicity assay showed that RAW264.7 cells exhibited greater cytotoxicity by SCGB3A2 + LPS compared to the individual treatments, demonstrating the critical role of SCGB3A2 as an LPS delivery molecule to macrophage cells as well (*Figure 4—figure supplement 4C*).

In LLC cells under SCGB3A2 + LPS, caspase-11 expression was upregulated in a diffused distribution pattern in the entire area and showed specific foci (*Figure 4I*). Importantly, the caspase-11 foci overlapped with incorporated LPS (*Figure 4I*). The expression of NLRP3 was also clearly up-regulated by LPS +SCGB3A2 and accumulated around the caspase-11 foci. We hypothesized that the incorporated LPS triggers formation of caspase-11 foci in LLC cells. As expected, when LPS was introduced into LLC cells using a DNA transfection reagent, LLC cells showed increased intracellular LPS signals and caspase-11 foci, overlapped with LPS (*Figure 4J*), confirming that the formation of caspase-11 foci is mediated by LPS introduction into the cytosol of LLC cells. We could not detect clear foci of caspase-1 in LLC cells, unlike the case of macrophages as previously reported (data not shown). Caspase-11 foci formation and NLRP3 upregulation driven by LPS + SCGB3A2 were also observed in RAW264.7 cells (*Figure 4—figure supplement 4D*). These results confirm that SCGB3A2 facilitates the delivery of LPS into the cytosol, in concert with the enhancement of non-canonical inflammasome signaling.

To confirm the importance of clathrin-mediated endocytosis of LPS via the SCGB3A2-SDC1 pathway for killing of LLC cells, the effect of clathrin inhibitor, Dynasore on the growth of LLC cells was examined in vitro (*Figure 4K–4M*). LLC cells had strong focal staining of SCGB3A2-HT and LPS$_{A448}$ at the corresponding locations to each other, while Dynasore potently inhibited the incorporation of SCGB3A2 and LPS into the cytosol of LLC cells (*Figure 4K*) and abrogated the activation of caspase-11 (*Figure 4L*). LDH release from LLC cells as indication for cytotoxicity was slightly upregulated by LPS +SCGB3A2, while this upregulation was not observed when cells were treated with either Dynasore or Wedelolactone (caspase-11 inhibitor) (*Kobori et al., 2004*) (*Figure 4M*, *Figure 4—figure supplement 5A*). Furthermore, when LLC-sh-casp-11 cells (*Figure 4—figure supplement 5B*) were subjected to the intravenous metastasis model with or without SCGB3A2 administration, they did not show any significantly different numbers of lung tumors after SCGB3A2 administration, in sharp contrast to the results with control LLC cells (*Figure 4N*). These results clearly indicate the importance of clathrin-mediated endocytosis of LPS +SCGB3A2 and caspase-11 activation for the killing of LLC cells in vivo.

## SCGB3A2-LPS promotes pytoptotic cell death of LLC cells

The SCGB3A2 + LPS complex promoted pyroptotic cell death morphology in cultured LLC cells (membrane swelling; *Figure 5A*). CCK8 assay confirmed the upregulation of pyroptotic cell death of LLC cells by essentially endotoxin-free SCGB3A2 plus a small amount of LPS (*Figure 5B*). Furthermore, flow cytometry analysis revealed the upregulation of propidium iodide (PI) positive cell death by SCGB3A2 + LPS (*Figure 5C*), demonstrating the formation of cell membrane pores, the characteristic feature of pyroptosis, induced by SCGB3A2 + LPS. Next, the induction of cell death by SCGB3A2 in vivo was examined in the LLC cell intravenous metastasis model (*Figure 5D*). Large necrotic areas were found in lung tumors from mice treated with early intravenous administration of SCGB3A2 (1$^{st}$ and 2$^{nd}$ week) (*Figure 5D*). Importantly, these necrotic areas showed enhanced expression of both caspase-11 and NLRP3, demonstrating that tumor cell death occurred through caspase-11-mediated non-canonical inflammasome activation (*Figure 5E and F*). These results clearly indicate that SCGB3A2 significantly promotes pyroptotic death of LLC cells both in vivo and in vitro.

## LLC-sh-SDC1 cells attenuate SCGB3A2-mediated inhibition of metastasis in the mouse LLC model

LLC-sh-SDC1 cells showed reduced susceptibility to the cytotoxic effects of LPS +SCGB3A2 complex in vitro (*Figure 6A*), accompanied by minimal enhancement of caspase-11 foci formation by

LPS +SCGB3A2 (*Figure 6B*, see *Figure 4I*). Heparin addition abrogated the increase of caspase-11 foci in LLC-sh-Control cells (*Figure 6B*), confirming the crucial role of heparin sulfate and SDC1 for caspase-11 foci formation. In vivo sensitivity of LLC-sh-SDC1 cells to SCGB3A2-mediated inhibition of metastasis was next analyzed. Tumor numbers in mice that received LLC-sh-SDC1 cells and SCGB3A2 were not significantly different from those that received LLC-sh-SDC1 cells and PBS, while tumor numbers with LLC-sh-Control cells were significantly reduced by SCGB3A2 co-injection, similar to that observed in *Figure 1* (*Figure 6C and D*). These experiments confirmed a pivotal role for SDC1 in SCGB3A2-mediated inhibition of LLC cell growth and metastasis in vivo. Next, to understand the reason for the differences in response to SCGB3A2 between LLC (susceptible) and B16F10 (resistant) cells, the baseline mRNA expression from inflammasome-related genes were examined (*Figure 6E*). As a result, *Casp11*, *Nlrp3*, *Aim2*, *Gsdmd, and Il1b* mRNAs were highly expressed only in LLC cells, suggesting that LLC cells have the machinery to activate a non-canonical inflammasome pathway driven by caspase-11 in combination with higher expression levels of cell surface SDC1 (see *Figure 3D and E*). Lastly, the effect of SCGB3A2 on the survival of lung-specific $Kras^{G12D}$ mutant mice was examined using $Kras^{G12D}$;Scgb3a2(fl/fl) and the littermate $Kras^{G12D}$;Scgb3a2(fl/+) mice (*Figure 6F*). Due to lung-specific activation of the $Kras^{G12D}$ allele, these mice developed lung cancer within 4 months of age. $Kras^{G12D}$;Scgb3a2(fl/fl) mice clearly showed a poorer survival rate than $Kras^{G12D}$;Scgb3a2(fl/+) mice. Based on these results, we propose a new model for SCGB3A2 delivery of LPS and activation of caspase-11(caspase-4) pathway via SDC1 receptor signaling, leading to pyroptosis of cancer cells (*Figure 6G*).

## Discussion

SCGB3A2 is a member of the secretoglobin family of proteins, which share a common four helical bundle subunit structure, exist as dimers, tetramers, and other oligomers, and some of which have also been implicated in tumor suppression (*Mukherjee et al., 2007*) without a clear understanding yet of the mechanistic pathway(s). This work takes a significant step forward to elucidate and describe a new pathway impacted by SCGB3A2 functioning as a tumor suppressor protein. Previously we showed that SCGB3A2 functions as an anti-inflammatory and anti-fibrotic agent in the lung (*Cai and Kimura, 2015*; *Cai et al., 2014*; *Chiba et al., 2006*; *Kido et al., 2014*; *Kurotani et al., 2011*; *Yoneda et al., 2016*). Because SCGB3A2 is mainly secreted by club cells in lung airways, it is reasonable to assume that a primary function of SCGB3A2 is to protect the hosts from pathogens and pathogen-associated molecular patterns such as LPS. The current study demonstrated that SCGB3A2 binds to and facilitates delivery of LPS into the cytosolic compartment through specific binding with SDC1, resulting in cell death via an inflammatory pathway leading to pyroptosis. This is commonly seen in the macrophage cell line RAW264.7, suggesting a possible conserved role for SCGB3A2 in host defense and enhancing the immune response through the non-canonical inflammasome pathway of pyroptosis. Notably, in the case of LLC cells, the uptake of SCGB3A2-sequestered small amount of LPS triggers inflammatory cell death, probably because of the abundant SDC1 expression on their cell surface. It is noteworthy that caspase-11 and human caspase-4/5 are specific to mammals (*Kayagaki et al., 2015*), while the SCGB superfamily of proteins, including SCGB3A2, have also evolved in mammalian lineages (*Jackson et al., 2011*), suggesting the co-emergent evolution as an 'input-output' for defense from invading pathogens.

SDC1 localization to uropods is functionally important as uropods accumulate growth factors and connect them at cell-to-cell contact points or junctions (*Børset et al., 2000*; *Yang et al., 2003*). Others demonstrated that the SDC1-specific HS sequence is important for targeting SDC1 to uropods (*Børset et al., 2000*). Our results that heparin treatment dramatically reduces SCGB3A2 and LPS binding and their incorporation into cells, as well as caspase-11 and NLRP3 induction suggest that SCGB3A2 appears to interact with the HS moiety of SDC1, which is concentrated in the membrane uropods.

Recently, it was reported that bacterial outer membrane vesicles (OMVs) deliver LPS into the host cell cytosol via clathrin mediated endocytosis (*Vanaja et al., 2016*). OMV is expected to work as a platform vaccine technology because of the potential to deliver small antigens and to modulate the immune system, however, it is highly toxic due to contamination with a large amount of LPS (*Acevedo et al., 2014*). In addition, guanylate-binding proteins (GBPs) are reported to have important function for interaction with cytosolic OMV and activation of caspase-11 (*Meunier et al., 2014*;

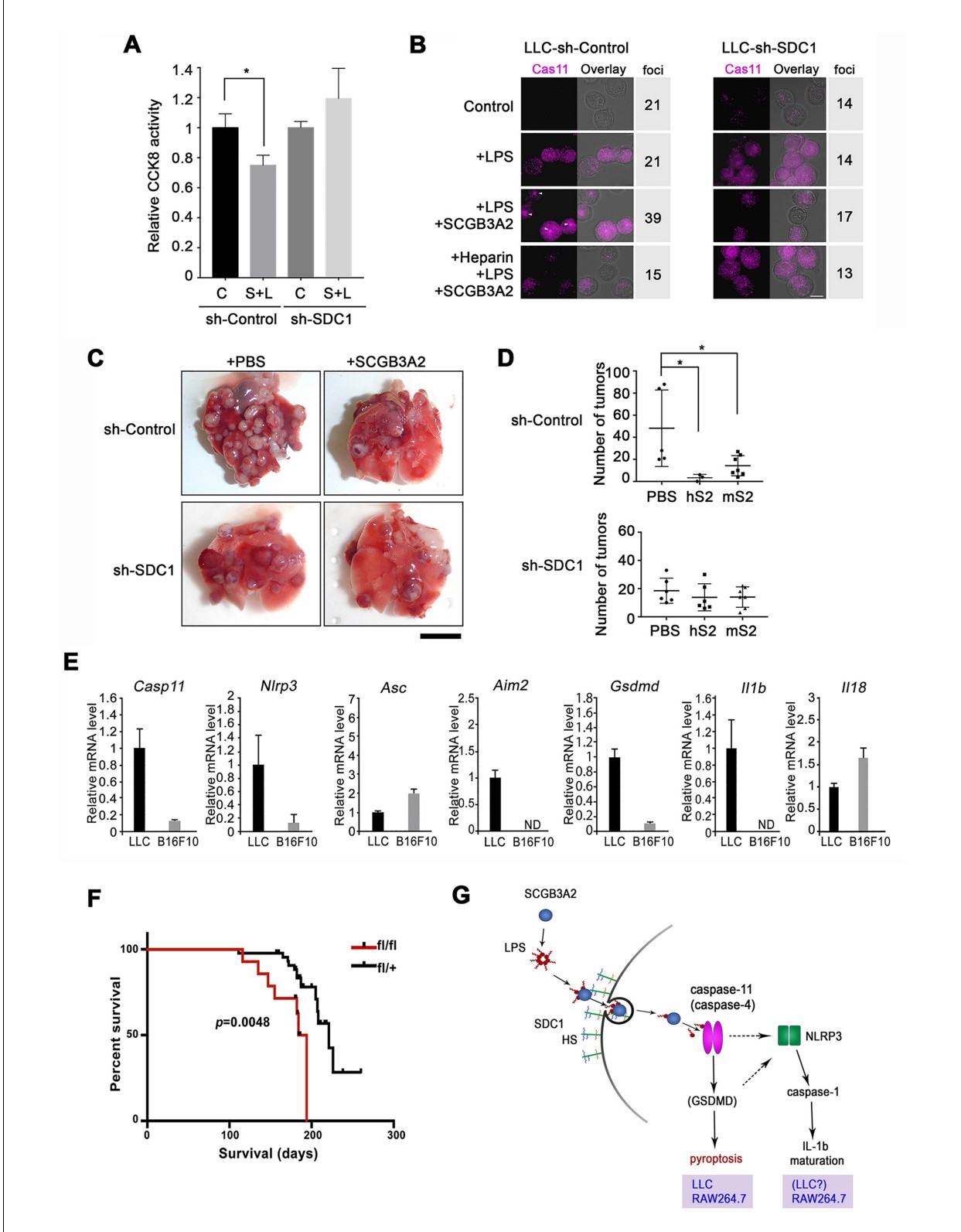

**Figure 6.** Evaluation for the requirement of SDC1 and other genes for SCGB3A2-LPS effect. (**A**) CCK8 assay using LLC-sh-Control and LLC-sh-SDC1 for 72 hr in 1% FBS-RPMI 1640 medium. C; control, S; human SCGB3A2 (200 ng/ml), L; LPS (O111:B4, 1 pg/ml). Averages ± SD from more than three independent experiments, each in triplicate are shown. *: p<0.05 by one-way ANOVA. (**B**) Immunofluorescent staining of caspase-11 and LPS$_{A594}$ using LLC-sh-Control and LLC-sh-SDC1 cells. Cells were maintained in 1% FBS-RPMI 1640 medium for 16 hr with or without human SCGB3A2 (1 μg/ml), LPS

*Figure 6 continued on next page*

*Figure 6 continued*

(1 µg/ml), and/or heparin (1 µg/ml). Arrowheads: caspase-11 foci. Bar = 10 µm. Numbers on the right indicate cells containing caspase-11 foci per a total 100 cells counted for each cell type. Data are the representatives of three independent experiments. (C) Intravenous metastasis model using LLC-sh-Control or LLC-sh-SDC1 cells. Representative lungs from the control (PBS) and SCGB3A2 groups that received treatment in the 1<sup>st</sup> week. N = 3–6 per group. Bar = 1 cm. (D) Summary of C for the numbers of pulmonary surface tumors in lungs of animals receiving treatment as indicated. hS2; human SCGB3A2, mS2; mouse SCGB3A2. A dot indicates a mouse. *p<0.05 by One-way ANOVA. (E) qPCR quantification of the relative mRNA expression levels for inflammasome related genes in LLC and B16F10 cells. Cells grown in 10%FBS-RPMI 1640 medium were harvested at 24 hr. Averages ± SD from more than three independent experiments, each in triplicate. ND; not detectable. (F) $Kras^{G12D}$-induced lung carcinogenesis survival curve for $Kras^{G12D}$;$Scgb3a2(fl/fl)$ (red line; fl/fl, n = 14) and littermate $Kras^{G12D}$;$Scgb3a2(fl/+)$ (black line; fl/+, n = 44) mice. (G) Schematic model for LPS entry into cells by SCGB3A2 through SDC1 receptor, leading to pyroptotic cell death.

DOI: https://doi.org/10.7554/eLife.37854.016

*Santos et al., 2018*). Our findings demonstrate that SCGB3A2 is incorporated into cytosol via a clathrin-mediated uptake mechanism, and that SCGB3A2 is a potent LPS disaggregation protein. Hence, SCGB3A2 might be an attractive protein which has a key function similar to OMV, but without any significant toxicity because of its natural occurrence and abundance in lung. It is also interesting to speculate that SCGB3A2 could liberate LPS from OMV to gain access to cell cytosols either from early endosome or from extra cellular spaces, collaborating with other host proteins such as GBPs. Whether this is the case requires further studies.

It was reported that the non-canonical inflammasome pathway governed by caspase-4/caspase-11, intrinsic to intestinal epithelial cells, plays a critical role in antimicrobial defense, causing pyroptotic cell death and shedding of infected cells (*Knodler et al., 2014*). These events could limit pathogen colonization of the intestinal epithelium. Likewise, it's conceivable that lung airway epithelial cells have an intrinsic non-canonical inflammasome pathway for antimicrobial defense, through the SCGB3A2 and SDC1 interaction. Moreover, the present results suggest that this non-canonical inflammasome pathway is retained in some cancer cells and this property could be used for cancer treatment. Importantly, it was reported that newborn *Sdc1(-/-)* mouse lungs show marked resistance against *P. aeruginosa* infection (*Park et al., 2001*). This study was extended to show the biological function of SDC1 in lung epithelial cells from a simple cell membrane receptor for growth factors and chemokines to that of modulating microbial pathogenesis and host defense (*Park et al., 2001*). The role of SCGB3A2 as a chaperon to deliver LPS to cell cytosols may initially be established to protect host cells from infection, while this mechanism may have evolved to protect host from cancer development by activation of the non-canonical inflammasome signaling pathway. Anti-tumor effects of endotoxin/LPS has been known for decades while the effects are still controversial; one reason is because the effects vary depending on different cancers (*Lundin and Checkoway, 2009*; *Ribi et al., 1983*). Our results could provide one of the reasons for the various sensitivities of different cancer cells to endotoxin.

Of note is that the levels of SDC1 expression differ depending on cancer types and are strikingly dysregulated in many cancer cells (*Akl et al., 2015*; *Teng et al., 2012*). Because loss of membranous SDC1 increases the mobility of cancer cells, resulting in enhancement of metastasis, in general, loss or weak expression of SDC1 in tumors is thought to be associated with unfavorable outcomes. In lung cancer patients, high serum levels of shed SDC1 and bFGF were associated with poor prognosis (*Joensuu et al., 2002*). Some reports also found cytoplasmic or nuclear localization patterns of SDC/HS in less differentiated malignant cells (*Akl et al., 2015*; *Burbach et al., 2003*; *Miyake et al., 2014*), however the underlying mechanism for this correlation is largely unknown. Cancer cells are notorious for changing/adapting in order to survive, such as acquisition of the resistance to chemotherapeutic reagents. In addition to the loss of contact with extracellular matrix, the various expression patterns (reduced, shed, or subcellular) of SDC1 in many malignant cancer cells might suggest that this could be one of their acquired properties; by losing the expression of SDC1 on their cell surface, they will become refractory to the microorganism/LPS triggering non-canonical inflammasome pathway, thus avoiding their own death. Further studies will be required, particularly regarding which cancer cell types possess the machinery and/or express the necessary genes and protein expression patterns that permit response via the non-canonical inflammasome pathway. Collectively, these findings could be utilized for the recognition of the importance of the inflammasome activation of cancer cells and the innate immune system for cancer targeting and treatment. The currently

available cancer immunotherapy is mainly targeted to the host immune cells, whereas our study shows the possibility to directly target the activation of non-canonical inflammasome pathway of cancer cells. Our findings may provide the new clue for the understanding of many cancers that are refractory to cancer immunotherapy mediated by immune cells. Combination of the cancer immunotherapy and the cancer cell self-destructive therapy could greatly advance the treatment of cancer patients.

# Materials and methods

### Key resources table

| Reagent type (species) Or resource | Designation | Source or reference | Identifiers | Additional information |
|---|---|---|---|---|
| Strain, strain background (M. musculus) | *Scgb3a2⁻/⁻* (C57BL/6N background) | In house | *Kido et al., 2014* | |
| Strain, strain background (M. musculus) | *Scgb3a2fl/fl* (C57BL/6N background) | In house | *Kido et al., 2014* | |
| Strain, strain background (M. musculus) | *Ccsp-Cre;LSL-K-rasG12D* (mixed background) | Francesco DeMayo (Baylor College of Medicine, now NIEHS) | *Moghaddam et al., 2009* | |
| Cell line (M. musculus) | LLC-Mhi | Glenn Merlino (NCI) | | Highly metastatic, MAP test negative |
| Cell line (M. musculus) | B16F10 | ATCC | CRL-6475 | |
| Cell line (M. musculus) | RAW264.7 | Raymond Donnelly (FDA) | | MAP test negative, authenticated by STR |
| Cell line (human) | ARH-77 ARH-77-mSDC1 | Ralph Sanderson (The University of Alabama at Birmingham) | | MAP test negative, authenticated by STR |
| Recombinant protein | mouse SCGB3A2 | NCI Protein Expression Core | *Kurotani et al., 2008* | |
| Recombinant protein | mouse SCGB3A2 | Hölzel Diagnostics | CSB-BP846028MO | |
| Recombinant protein | mouse SCGB3A2 | CosmoBio USA | CSB-EP846028MO | |
| Recombinant protein | human SCGB3A2 | In house (APCBio Innovations) | *Cai et al., 2014* | |
| Antibody | anti-mouse SCGB3A2 (rabbit polyclonal) | E. Coli expressed mature SCGB3A2 in vector pET-32a(+) used as an antigen to produce a polyclonal antibody (produced by Macromolecular Resources, Fort Collins, CO) | *Niimi et al., 2001* | 1:5000 for IHC |

*Continued on next page*

*Continued*

| Reagent type (species) Or resource | Designation | Source or reference | Identifiers | Additional information |
|---|---|---|---|---|
| Antibody | anti-human SCGB3A2 (polyclonal) | In house (APCBIo Innovations) | *Cai et al., 2014* | 1:1000 for WB |
| Antibody | anti-mouse/human SDC1 | Pynog W Park (Harvard Med Sch) | | 1:200 for IHC/IF 1:1000 for WB |
| Antibody | PE-rat anti-mouse SDC1 | BD Pharmingnen | clone 281.2 | 1:200 for FACS |
| Antibody | anti-clathrin heavy chain | Cell Signaling Technology | P1663 | 1:200 for IF |
| Antibody | anti-EEA1 (monoclonal) | Cell Signaling Technology | C45B10 | 1:200 for IF |
| Antibody | anti-IL-1β (monoclonal) | Cell Signaling Technology | Clone 3A6 | 1:1000 for WB |
| Antibody | anti-caspace-11 (monoclonal) | Thermo Fisher Scientific | Clone 17D9 | 1:1000 for WB 1:100 for IHC/IF |
| Antibody | anti-ICAM1 (monoclonal) | Thermo Fisher Scientific | MA5407 | 1:200 for IF |
| Antibody | anti-caspace-1 (monoclonal) | Thermo Fisher Scientific | Clone 5B10 | 1:1000 for WB 1:100 for IF |
| Antibody | anti-NLRP3 (monoclonal) | Adipogen | AG-20B-0014 | 1:1000 for WB 1:200 for IHC/IF |
| Antibody | anti-LAMP1 (monoclonal) | Santa Cruz | sc-17768 | 1:200 for IF |
| Antibody | anti-HaloTag (polyclonal) | Promega | G9281 | 1:100 for IF 1:1000 for WB |
| Antibody | anti-GAPDH (monoclonal) | Proteintech | 60004–1-Ig | 1:5000 for WB |
| Antibody | anti-LPS (monoclonal) | Abcam | ab35654 | 1:1000 for WB |
| Chemical compound, drug | LPS from *E. coli* O111:B4 | Sigma-Aldrich | L4391 | |
| Chemical compound, drug | Ra mutant LPS from *E.coli* EH-100 | Sigma-Aldrich | L9641 | |
| Chemical compound, drug | LPS from *E.coli* K-235 | Sigma-Aldrich | L2018 | |
| Chemical compound, drug | LPS from *Salmonella typhimurium* | Sigma-Aldrich | L2262 | |
| Chemical compound, drug | heparin sodium salt from porcine intestinal mucosa | Sigma-Aldrich | H3393 | |
| Chemical compound, drug | imidazole | Sigma-Aldrich | I5513 | |
| Chemical compound, drug | zinc sulfate solution | Sigma-Aldrich | Z2876 | |
| Chemical compound, drug | Dynasore | AdooQ Bioscien | A12726 | |
| Chemical compound, drug | Wedelolactone | AdooQ Bioscien | A14804 | |
| Chemical compound, drug | LPS-EB Biotin | InvivoGen | tlrl-3blps | |

## Cell culture

The LLC cells used in this study were the LLC-Mhi cell line (obtained from Dr. Glenn Merlino, NCI), which is a high metastatic subline derived from LLC tumors described previously (*Day et al., 2012*). B16F10 cells were purchased from American Type Culture Collection. ARH-77 and ARH-77-mSDC1 cells were kindly provided by Dr. Ralph D. Sanderson (University of Alabama at Birmingham), and

RAW264.7 cells by Dr. Raymond P. Donnelly (FDA). LLC, ARH-77, and RAW264.7 cells were all tested negative for mycoplasma (NCI Core facility) and authenticated by STR analysis (IDEXX BioResearch). LLC and B16F10 cells were cultured in RPMI 1640 Medium (LONZA) with heat-inactivated fetal bovine serum (FBS), supplemented with penicillin/streptomycin (1:100) at 37 °C, 5% $CO_2$. Culture of LLC cells was carried out under various concentrations of FBS, as indicated in the Figure legends. For LPS stimulation, RAW264.7 cells were cultured in OPTI-MEM™ I reduced serum medium (Thermo Fisher Scientific) for times indicated in the text. LPS transfection was performed using X-tremeGene HP DNA transfection reagent (Roche Applied Science).

## Protein microarray

SCGB3A2 binding proteins were identified using Protoarray™ Human Protein Microarray v5.0 Protein-Protein Interaction Kit for biotinylated proteins (Thermo Fisher Scientific, PAH0525101, >9000 proteins included). Experiments were carried out according to procedures provided by the manufacturer. First, a biotin label was introduced into recombinant human SCGB3A2 protein using Biotin-XX Microscale Protein Labeling Kit (Thermo Fisher Scientific B30010), which was then used to probe Protoarray Human protein microarrays. The microarrays were washed with washing buffer (PBS containing 10% Synthetic Block (included in the kit) and 0.1% Tween 20 (Thermo Fisher Scientific)), and probed with Alexa Fluor 647 conjugated streptavidin (included in the kit). After washing, the microarrays were dried and scanned by a fluorescent microarray scanner (Perkin Elmer, Scanarray Express) to obtain the data. Software for the data analysis (Protoarray Prospector) was also provided by the manufacturer.

## RNA interference by retrovirus-based shRNA

The shRNA constructs were purchased from transOMIC for mouse SDC1, from ORIGENE for mouse TLR4 and mouse caspase-11. Retroviral constructs were transfected into Phoenix packaging cells by using X-tremeGene HP DNA transfection reagent (Roche Applied Science). Drug selection and cell cloning were conducted in the presence of 2 μg/ml puromycin by the limited dilution method. shRNA constructs used for mouse Sdc1 knock down are as follows: pMLP-Sdc1-sh1; 5'-CGGGGA TGACTCTGACAACTTA-3', 5'-TAGTGAAGCCACAGATGTA-3', and 5'-TAAGTTGTCAGAGTCA TCCCCA-3', pMLP-Sdc1-sh2; 5'-ACAGGCAGCTGTCACATCTCAA-3', 5'-TAGTGAAGCCACAGATG TA-3', and 5'-TTGAGATGTGACAGCTGCCTGG-3'), and pMLP-Sdc1-sh3; 5'-CCAAGACTTCACC TTTGAAACA-3', 5'-TAGTGAAGCCACAGATGTA-3', and 5'-TGTTTCAAAGGTGAAGTCTTGT-3'. shRNA sequences used for mouse TLR4 knock down are as follows: 5'-CACTTAGACCTCAGCTTCAA TGGTGCCAT-3' and 5'-TGCCTTCACTACAGAGACTTTATTCCTGG-3'. shRNA sequences used for mouse *Casp11* knockdown are as follows: 5'-TAACAATGCTGAACGCAGTGACAAGCGTT-3', 5'-ACAGCACATTCCTGGTGCTAATGTCTCAT-3' and 5'-ATATTCCTGAAGGTGCAACAATCATTTGA-3'.

## Co-immunoprecipitation assay

COS-1 cells were transfected with 2.5 μg each of candidate gene cloned into pcDNA3.1/Myc-His vector, the human SCGB3A2 (NM_054023) open reading frame cloned into pcDNA3.1 with a C-terminal FLAG tag, or a control plasmid by using X-tremeGene HP DNA transfection reagent. Both cells and media were harvested 48 hr after transfection. The culture media containing cells were centrifuged at 500 *g* for 10 min at 4°C and the supernatant was collected (media supernatant). Cells were lysed in 400 μL CHAPS IP buffer-1 (1% CHAPS, 150 mM NaCl, 50 mM Tris-HCl, pH 7.4, protease inhibitor complete-mini 1 tablet/10 ml) and sonicated two times for 5 s each on ice. The cell lysates were centrifuged at 15,000 g for 10 min at 4°C and the supernatant was collected (cell lysate supernatant). The media supernatant and cell lysate supernatant were combined, which were pre-cleared with Protein G-Agarose (Santa Cruz Biotechnology) at 4°C for 3 hr, followed by incubation with FLAG-tagged gel (20 μL; #3326, MBL) at 4°C overnight. The gel-immunocomplexes were washed twice with CHAPS IP buffer-2 (0.1% CHAPS, 500 mM NaCl, 50 mM Tris-HCl, pH 7.4) for 20 min each and then washed twice with CHAPS IP buffer-3 (0.1% CHAPS, 50 mM Tris-HCl, pH 7.4) for 20 min each.

Immunoprecipitated samples were separated by SDS-PAGE and electroblotted to PVDF membranes. Blocking was carried out with 5% skim milk in TBST (Tris-buffered saline; Tris-HCl, pH

7.4 + 0.1% Tween 20) and the membrane was subsequently incubated with anti-Myc mouse monoclonal antibody (1:1000, 9B11, Cell signaling) at 4°C overnight followed by incubation with sheep anti-mouse IgG HRP-linked F(ab')$_2$ fragment (1:2000; NA9310, GE Healthcare). Signals were detected as described for western blotting.

## Streptavidin pull down assay

LPS-Biotin (1 mg/ml) and immobilized Streptavidin agarose gel were incubated for 30 min at 4 °C, and after biotin blocking, 1.25 mg/ml recombinant human SCGB3A2 was added as a pray protein and incubated for 1 hr at 4 °C. Ten % of flow through was used as an input. After washing several times, the gel was boiled for 5 min with SDS sample buffer and the supernatant was used for western blotting.

## Imidazole-zinc staining

Imidazole-zinc staining was carried out as previously reported (*Rodríguez and Hardy, 2015*). Briefly, LPS dissolved and/or SCGB3A2 diluted in water were loaded onto 0.8% agarose gel in full in a well to make sure the content reaching to gel surface and run at 50V in TAE (Tris-acetate-EDTA; 40 mM Tris, 20 mM acetic acid, and 1 mM EDTA, pH 8.0) buffer until dye reached to the gel bottom. The gel was washed with ddH$_2$O and immersed in 0.2 M imidazole for 20 min with gentle agitation. After discarding solution and washing with ddH$_2$O, the gel was placed in the dark and incubated with 0.3 N zinc sulfate solution for several minutes. Then the gel was rinsed with ddH$_2$O to stop staining and an image was taken with ChemiDoc$^{TM}$ imaging system (Bio-Rad). For double staining experiments, the gel was stained with 0.25% Coomassie Brilliant Blue solution after the gel image of Imidazole-zinc staining was scanned.

## Quantitative RT-PCR

Total RNA was extracted by TRIzol (Life Technologies) and reverse transcribed into cDNA by using SuperScript III reverse transcriptase (Life Technologies) according to the manufacturer's protocol. Analysis of mRNA levels was performed on a 7900 Fast Real-Time PCR System (Life technologies) with SYBR Green-based real-time PCR. The primer sequences used for real-time PCR are as follows:

(sense) 5'-CTCAGAGCCTTTTGGACAGG-3' and
(antisense) 5'TACAGCATGAAAGCCACCAG-3' for mouse *Sdc1*;
(sense) 5-TGTGTACACGGAGAAACATTCAG-3 and
(antisense) 5- GCAAAGAGAAAGCCGATCAC −3 for mouse *Sdc2*;
(sense) 5-AACTGAGGTCTTGGCAGCTC-3' and
(antisense) 5'-TACACCAGCAGCAGGATCAG-3' for mouse *Sdc4*;
(sense) 5'-CCAATTTTTCAGAACTTCAGTGG-3' and
(antisense) 5'-AGAGGTGGTGTAAGCCATGC 3' for mouse *Tlr4*;
(sense) 5'-GCTGATGCTGTCAAGCTGAG-3' and
(antisense) 5'-GAGCCTCCTGTTTTGTCTCG-3' for mouse *Casp11*;
(sense) 5'-CCTCTGTGAGGTGCTGAAAC-3' and
(antisense) 5'-TCAGGCTTTTCTTCCTGGAG-3' for mouse *Nlrp3*;
(sense) 5'-TGGGCTGTTTAAAGTCCAGAAG-3' and
(antisense) 5'-TTTGTTTTGCTTGGGTTTCC3' for mouse *Aim2*;
(sense) 5'-ACATGGGCTTACAGGAGCTG-3' and
(antisense) 5'-ACTCTGAGCAGGGACACTGG-3' for mouse *Asc*;
(sense) 5'-TGTCTGGTGCTTGACTCTGG-3' and
(antisense) 5'-CTGGGTTTCACTCAGCACAG-3' for mouse *Gsdmd*;
(sense) 5'-GCTGTGACCCTCTCTGTGAAG-3' and
(antisense) 5'-TTTCAGGTGGATCCATTTCC-3' for mouse *Il18*;
(sense) 5'-AAAGCTCTCCACCTCAATGG-3' and
(antisense) 5'-AGGCCACAGGTATTTTGTCG-3' for mouse *Il1b*;
(sense) 5'- ACAAGACCCACGTGGAGAAG −3'.

## Western blotting

Cells were lysed in RIPA lysis buffer (Millipore) and protein concentration was measured by BCA protein assay kit (Thermo Fisher Scientific). Samples were separated by SDS-PAGE and electroblotted to polyvinylidene fluoride (PVDF) membranes (GE Healthcare). In the case of SDC1 detection, cell membrane extract was prepared using Subcellular Protein Fractionation Kit for Cultured Cells (Thermo Fisher Scientific) according to the manufacturer's protocol, and blotted to cationic nylon membrane (Immobilon Ny; Millipore). Signals were visualized with SuperSignal West Dura Chemiluminescent Substrate (Thermo Fisher Scientific) according to the manufacturer's protocol. Chemiluminescence was quantitated using a Bio-Rad ChemiDoc$^{TM}$ MP imaging system.

## FACS analysis

For LPS and cell binding assay, cells were washed with PBS and incubated with Alexa 488 or 594-conjugated LPS from *E.coli* 055:B4 (L-23351 or L-22353, 1 µg) (Thermo Fisher Scientific) with or without SCGB3A2 (1 µg/ml) at 4 ℃ for 30 min. After washing with PBS, the cells were analyzed in a FACS Canto II (Becton Dickinson). For the SCGB3A2 and LLC cell binding assay, LLC cells were incubated with recombinant mouse or human SCGB3A2, washed with PBS, incubated with anti-SCGB3A2 antibody for 30 min followed by PE-rabbit IgG secondary antibody for 30 min. For SDC1 expression analysis, LLC cells were harvested in PBS and stained with PE-rat anti-mouse SDC1 (clone 281.2, BD Pharmingnen) for 30 min at 4 ℃. For Annexin V/PI analysis, Dead Cell Apoptosis Kit with Annexin V FITC and PI, for flow cytometry (V13242, Thermofisher Scientific) was used. Cells were harvested using a scraper and washed with cold PBS and stained with Annexin V-Alexa 488 and PI in 1x Annexin binding buffer for 15 min. As a compensation control, FITC-stained only or PI-stained only cells were prepared by inducing cell death by incubation in 70% EtOH for 10 min. All experiments were carried out in the NCI Flow Cytometry Core Facility.

## LLC cells mouse metastasis model

LLC cells ($2 \times 10^5$ cells) were intravenously administered to C57BL/6N mice (Charles River, Frederick, MD), followed by daily intravenous administration of recombinant mouse or human SCGB3A2 (0.25 mg/kg/day) for 7 days starting at day 0 (30 min after LLC cells injection), 7, or 14 or during the entire experimental period of 20 days, or PBS injection for 20 days as control. Mice were killed on day 21 and the numbers of lung metastasized tumors evaluated. Some lungs were subjected to histological analysis. *Scgb3a2(-/-)* mice(*Kido et al., 2014*) used in the metastasis model were those 10 times backcrossed to C57BL/6N, and the littermates wild-type mice were used as control.

## Lung carcinogenesis study

*Ccsp-Cre;LSL-Kras$^{G12D}$* conditional mutant mice on the 129SvJ-C57BL/6 mixed background (*Jackson et al., 2001*; *Moghaddam et al., 2009*) which express the oncogenic *Kras$^{G12D}$* gene in lung-specific fashion were provided by Francesco DeMayo (Baylor College of Medicine, Houston, TX). *Scgb3a2(fl/fl)* mice, previously described (*Kido et al., 2014*), were backcrossed to C57BL/6N mice three times. *Ccsp-Cre;LSL-Kras$^{G12D}$* and *Scgb3a2(fl/fl)* mice were crossed to produce *Ccsp-Cre; LSL-Kras$^{G12D}$;Scgb3a2(fl/fl)* (tentatively named *Kras$^{G12D}$;Scgb3a2(fl/fl)*) and littermate *Ccsp-Cre;LSL-Kras$^{G12D}$;Scgb3(fl/+)* (tentatively named *Kras$^{G12D}$;Scgb3a2(fl/+)*) mice, and male mice were used in the study. Mice were maintained under standard specific-pathogen-free conditions, and the studies were carried out according to the guidelines for animal use issued by the National Institutes of Health and after approval by the National Cancer Institute (NCI) Animal Care and Use Committee.

## HaloTag imaging

To construct a HaloTag-mouse SCGB3A2 (mSCGB3A2-HT) expression vector, pFC14A HaloTag CMV Flexi Vector (Promega) was fused to C-terminal of mouse SCGB3A2 cDNA. Primers for the SCGB3A2 HaloTag plasmid were designed using the Flexi Vector Primer Design Tool web site. A HaloTag Coding Region Control Expression Vector (Control-HT) was designed according to the manufacture's instruction. mSCGB3A2-HT or Control-HT was transfected to HEK293 cells using X-tremeGENE HP DNA Transfection Reagent and after 48 or 72 hr, supernatant was collected and concentrated with Amicon Ultra (Millipore) and stored at −80 ℃ until use. The transfection efficiency was confirmed with microscopy using HaloTag TMRDirect ligand. For uptake of HT-mSCGB3A2 into

LLC cells, after addition of HT-mSCGB3A2, cells were stained with HaloTag TMR ligand for short incubation time or HaloTag TMRDirect ligand overnight. After two washes with PBS, the cells were visualized under a microscope.

## Histological analysis

Lung samples were fixed in 10% buffered formalin under 20 cm $H_2O$ pressure, embedded in paraffin, sectioned at 4 μm by microtome and performed with Hematoxylin and Eosin staining (H & E).

## TUNEL assay

Terminal deoxynucleotidyl transferase-mediated dUTP-biotin nick end labeling (TUNEL) analysis was performed using DeadEnd Fluorometric TUNEL System (G3250, Promega) according to the manufacturer's instructions. Total tumor areas and TUNEL positive areas were measured using imageJ software, and a percentage of TUNEL positive areas per total tumor areas was calculated

## Immunofluorescence analysis

Cells were seeded on glass coverslips (Nunc Lab-Tek Chambered Coverglass (15583PK, Nunc). After fixation with 10% buffered formalin for 10 min at room temperature (RT), cells were permeabilized with 100% MeOH at −20℃ for 10 min. Blocking was done with 1% BSA in PBS for 1 hr and cells were stained with primary antibodies for 1 hr at RT. After wash with PBS, cells were stained with secondary antibodies (1:200, Alexa flour, Molecular Probe) for 45 min at RT. Stained signals were analyzed under confocal microscope (Zeiss 510/710) according to the NCI confocal microscope facility manual or Keyence microscope BZ-X700.

## SCGB3A2 modeling

A SCGB3A2 dimer model was build starting from a consensus secondary structure prediction obtained using several procedures including I-TASSER (https://zhanglab.ccmb.med.umich.edu/I-TASSER/); LOMETS (https://zhanglab.ccmb.med.umich.edu/LOMETS/); RaptorX (http://raptorx.uchicago.edu); Swissmodel (https://swissmodel.expasy.org); Phyre2 (http://www.sbg.bio.ic.ac.uk/phyre2); BHAGEERATH-H (http://www.scfbio-iitd.res.in/bhageerath/bhageerath_h.jsp) and Quark (https://zhanglab.ccmb.med.umich.edu/QUARK/). The above-mentioned procedures were used as available in their respective web-site implementations as of March 2017. The methods explored span the spectrum of structure prediction techniques including threading, library-based methods, etc. None of the methods explored produced a compact structure. The helical motifs were properly identified by all models. The consensus helical regions as described in *Figure 4—figure supplement 2B* were manually aligned against the uteroglobin structure (PDB ID:1UTG) identified as the closest homolog of SCGB3A2 for which an experimental structure is currently available. The missing sections connecting the helical motifs were modeled as loops to the sole purpose of connecting the helices in an initial workable model. The model was then refined using Feedback Restrain Molecular Dynamics (FRMD). FRMD is based on a self-consistent procedure to bias molecular dynamics trajectories towards a refined conformation using experimental information from multiple sources including X-ray diffraction or NMR data when available (*Cachau, 1994*; *Cachau et al., 1994*; *González-Sapienza and Cachau, 2003*). The procedure is conceptually similar to a reversed molecular replacement protocol when using X-ray data, with the additional advantage that only those regions of the molecule in agreement with the crystallographic data are affected by the crystallographic constrain, as weighted by the FRMD protocol thus preserving the structural homology when available (*Cachau et al., 1994*). FRMD was implemented in QMRx (*Fadel et al., 2015*) using X-plor-NIH (*Schwieters et al., 2003*) to compute the crystallographic restrains and GROMACS 5.1.4 (*Abraham et al., 2015*) to drive the molecular dynamics (MD) calculations using the Amber ff99sb-ildn force field for all MD calculations. All calculations were performed using a time step of 2 fs. All bonds were constrained for all MD calculations. The leapfrog algorithm was used for integration using a velocity rescaling thermostat (Noose-Hover) with a 0.1 ps coupling constant. Electrostatic forces were computed using a distance criteria, and a cutoff of 10 Å was used for van der Waals interactions. No periodic boundary conditions were used aside from the periodicity resulting from the X-ray constrains. The system was freely equilibrated at T = 300 K for 5 ns without constrains, the purpose of this short run was to relax the initial model without losing the original shape of the

model. The model was then fully relaxed using FRMD with X-ray restrains as described in (Cachau et al., 1994) and Fcalc values computed for PDB ID: 1UTG in-lieu of experimental values not deposited for this entry in the Protein Data Bank, and limited to a 6 Å resolution cutoff. The nature of the FRMD procedure restricts the value of energy-based monitors. The convergence of the model was monitored using a crystallographic R factor and RMSD (root mean square deviation) against the reference structure for homologous residues (see *Figure 4—figure supplement 2B*). The trajectory converges to the structure shown in *Figure 4—figure supplement 2* after 350 ns with an R value of 9.3 (6 Å) and RMSD 3.2 Å. The MD trajectory was continued for another 350 ns without noticeable changes in the structure. The dimer structure was used to explore possible tetrameric arrangements by rolling a dimer against another using GROMACS and the AMBER force field to probe the interaction. A favorable arrangement was detected as described in *Figure 4—figure supplement 2F*. The number and placement of Cys in 1UTG and SCGB3A2 are different. Thus, SCGB3A2 was modeled replacing Cys 48 by Ala to avoid the possible bias that could have resulted from imposing a disulfide bond during the MD calculation. Ala 48 was then replaced back to Cys in the final dimer model where the two Cys S atoms appear at less than 2.5A from each other suggesting a proper placement of the Cys 48 in the dimer. FMRD can be used to estimate the data lost during the modeling procedure by reversing the refinement procedure that is 1UTG was modeled from the final model of SCGB3A2 using an identical protocol as previously used to model SCGB3A2 from 1UTG. The structure of 1UTG thus modeled agrees with the experimental one with an RMSD 3.5 Å (backbone atoms).

## DLS

Dynamic light scattering analysis (DLS) was performed using DynaPro Nanostar (Wyatt). The radii of LPS, SCGB3A2, and LPS-SCGB3A2 complex were determined after samples were centrifuged and dissolved in 50 µL of 0.22 µm filtered sterile PBS. The evaluation of data was performed by Dynamics V7 software.

## *Limulus Amebocyte* lysate (LAL endotoxin) assay

LPS quantification in each SCGB3A2 recombinant protein was performed using the ToxinSensor™ Chromogenic LAL Endotoxin Assay Kit (L00350, GenScript).

## LDH assay

Cells grown in 96 flat bottom well plates were incubated with or without SCGB3A2 and/or LPS (O111:B4) in the media for indicated times as described in the figure legends. Cell supernatants were evaluated for the presence of cytoplasmic enzyme lactate dehydrogenase (LDH) using the Pierce LDH Cytotoxicity Assay Kit (Thermo Fisher Scientific). Cytotoxicity was calculated according to the kit instructions; as a percentage of (experimental LDH − spontaneous LDH)/(maximum LDH release − spontaneous LDH).

## Statistical analysis

Statistical analysis was carried out using GraphPad Prism v7. Data are shown as means ± SD. Levels of significance for comparison between samples were determined by student's *t*-test or one-way ANOVA. For the lung carcinogenesis study, the Kaplan-Meier method was used to estimate survival rates of mice and the log-rank (Mantel-Cox) test for comparing survival differences between groups. P values of < 0.05 were considered statistically significant.

## Acknowledgements

We are grateful to the following people; Pyong W Park (Harvard Medical School) for SDC1 antibodies and technical advice, Ralph D Sanderson (University of Alabama at Birmingham) for ARH-77 and ARH-77-mSDC1 cells, Yoshihiko Yamada (NIDCR) and William K Gillette (Frederick National Laboratory of Cancer Research) for their advice, John Buckley and Yoshinori Takizawa for technical support, Frank J Gonzalez (NCI) for critical review of the manuscript, and Karen M Wolcott (NCI Flow Cytometry Core Facility), Susan Garfield (NCI Confocal Microscopy Core Facility), and Grzegorz Piszczek (NHLBI Biophysics Core Facility for DLS analysis) for their help in carrying out various

experiments. This work was supported in part with Federal funds from the Frederick National Laboratory for Cancer Research, National Institutes of Health, under contract HHSN261200800001E. The content of this publication does not necessarily reflect the views or policies of the Department of Health and Human Services, nor does mention of trade names, commercial products or organizations imply endorsement by the U.S. Government.

## Additional information

### Competing interests

Aprile L Pilon: ALP is an employee of APCBio Innovations and has a 51% ownership interest in APCBio Innovations, which has an interest in commercializing the rhSCGB3A2. US Patent Application (No. 62/619,511) was filed for the work described in this manuscript. The other authors declare that no competing interests exist.

### Funding

| Funder | Grant reference number | Author |
| --- | --- | --- |
| National Cancer Institute | ZIA BC 010449 | Shioko Kimura |

The funders had no role in study design, data collection and interpretation, or the decision to submit the work for publication.

### Author contributions

Shigetoshi Yokoyama, Conceptualization, Validation, Investigation, Visualization, Methodology, Writing-original draft, Writing—review and editing; Yan Cai, Miyuki Murata, Takeshi Tomita, Mitsuhiro Yoneda, Validation, Investigation, Methodology, Writing—review and editing; Lei Xu, Formal analysis, Investigation, Writing—review and editing; Aprile L Pilon, Resources, Writing—review and editing, Provided recombinant protein with the quality that cannot be commercially available, and that played a crucial role in our study to obtain the results; Raul E Cachau, Resources, Funding acquisition, Validation, Investigation, Visualization, Methodology, Writing-part of original draft, Writing-review and editing; Shioko Kimura, Conceptualization, Resources, Supervision, Funding acquisition, Validation, Writing-original draft, Writing-review and editing

### Author ORCIDs

Shigetoshi Yokoyama http://orcid.org/0000-0003-4175-0548
Shioko Kimura http://orcid.org/0000-0001-9627-6818

### Ethics

Animal experimentation: This study was performed in strict accordance with the recommendations in the Guide for the Care and Use of Laboratory Animals of the National Institutes of Health. All animals were housed in a temperature and humidity controlled specific pathogen-free facility under a 12-hour light/dark cycle with free access to water and food, and handled in a humane manner in an AAALAC-accredited facility in accordance with the established NIH Guidelines. Animal studies were carried out under protocols approved by the National Cancer Institute Animal Care and Use Committee (Protocol number: LM-091).

### Decision letter and Author response

Decision letter https://doi.org/10.7554/eLife.37854.022
Author response https://doi.org/10.7554/eLife.37854.023

## Additional files

### Supplementary files

• Supplementary file 1. LPS contents in various preparations/batches of recombinant SCGB3A2

DOI: https://doi.org/10.7554/eLife.37854.017

• Supplementary file 2. 116 proteins identified as those significantly bound to human SCGB3A2 in protein array. Proteins with Z-Factor >0.9, Z-Score >4.0 are listed. High score indicates a strong protein-protein interaction.
DOI: https://doi.org/10.7554/eLife.37854.018

• Supplementary file 3. SCGB3A2 candidate receptors identified by human protein interaction array
DOI: https://doi.org/10.7554/eLife.37854.019

• Transparent reporting form
DOI: https://doi.org/10.7554/eLife.37854.020

## Data availability

All data generated or analysed during this study are included in the manuscript and supporting files. Source data files have been provided in Supplementary table 2 and 3.

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
