## [Decision Letter]

Thank you for submitting your article "Secretoglobin 3A2 promotes cellular uptake of LPS through syndecan-1 leading to pyroptotic cell death" for consideration by *eLife*. Your article has been reviewed by three peer reviewers, and the evaluation has been overseen by a Reviewing Editor and Michel Nussenzweig as the Senior Editor. The following individual involved in review of your submission has agreed to reveal his identity: Seth L Masters (Reviewer #2).

The reviewers have discussed the reviews with one another and the Reviewing Editor has drafted this decision to help you prepare a revised submission. Whereas all of the essential revisions must be addressed, we realize that the Essential Revision comment is a considerable request, but we urge you to consider addressing this one as it is an important one.

Summary:

This paper describes a novel concept in which the secretoglobin SCGB3A2 binds LPS and the cell surface receptor syndecan-1 (SDC1). This allows LPS entry into cancer cells. This then activates pyroptosis and kills cancer cells via the non-canonical inflammasome, independent of TLR4.

Strengths:

a) In this paper an unbiased approach was taken, relating to endotoxin contamination of SCGB3A2 protein preparations. This was followed by selected screening for the receptors mediating entry into the cell. This led to identification of SDC1 receptor as the receptor that facilitates LPS entry into cells.

b) The authors use both biochemical and genetic approaches to conclude that SCGB3A2 binds to LPS, facilitating LPS uptake through the SDC1 receptor, leading to caspase-11-dependent pyroptosis.

c) The authors data have relevance for our understanding of the mechanism of entry of LPS into cells, and for the normal function of secretoglobin family 3A member 2 (SCGB3A2). These data also show that this pathway may trigger pyroptosis.

Essential revisions:

1) The main claim, that this pathway triggers pyroptosis, remains unproven at the genetic level. Although the authors indicate that LLC cells undergo pyroptosis following treatment with LPS and SCGB3A2 (Figure 5), the induction of cell death is low (Figure 5B and C). In addition, there is no evidence provided that supports the conclusion that death is caspase-11-dependent cell death (Figure 6F); this is a critical issue. Are *Casp11* KO or *Gsdmd* KO LLC gRNA cells indeed resistant to LPS-induced cell death? Genetic deletion or depletion of caspase-4/5 or caspase-11 needs to be performed to confirm that the effect observed is pyroptosis due to LPS triggering the non-canonical inflammasome. The authors should also make an effort to assess whether inhibiting caspase-11 activation prevent pyroptosis and cell death. Finally, the authors should show that inhibition of clathrin mediated endocytosis (by Dynasore) inhibits killing of LLC. These experiments would add much-needed support to the overall premise outlined by the authors. As a side note, more standard assays for pyroptosis (such as LDH release) should be done.

2) Does LPS given on its own suppress tumor growth in *Scgb3a2(-/-)* mice? How much of the tumor suppressive effect in Figure 1 is due to the LPS in the recombinant protein? For example, if SCGB3A2 suppresses tumors in mice how do SDC1 low tumors behave? Does knocking down or CRISPRing out SDC1 from LLC rescue them from LPS toxicity in vivo? The authors should CRISPR out TLR4 and SDC1 from LLC cells engraft them into their mouse model measure tumor load with and without LPS+SMGB3A2.

3) The authors concluded that NLRP3 mediates pyroptosis downstream of caspase-11 (Figure 6F). However, this contradicts the field's consensus that NLRP3 is dispensable for pyroptosis in non-canonical inflammasome signaling (Sharma and Kanneganti, 2016). This study does not address this contradiction. The authors need to address this discrepancy or they may wish to simply remove discussion of NLRP3, as it is highly unlikely that NLRP3 is involved in their system.

4) The binding of LPS-SCGB3A2 was examined by agarose gel separation/reverse staining and DLS assay; the shift of LPS aggregate to lower sizes was shown (Figure 2B, C, and D). Given these findings, the authors concluded that LPS binds to SCGB3A. However, these shifts do not directly support the binding of LPS to SCGB3A2. For example, they may imply that SCGB3A has some LPS dissolution activity. The LPS-SCGB3A2 binding study should be re-examined by standard assays such as Surface Plasmon Resonance analysis and/or biotin-LPS pulldown. As a side note, the presented DLS assay is missing a SCGB3A alone control (Figure 2D).

5) In the Materials and methods section, the procedure for recombinant SCGB3A protein preparation was not disclosed. This is of particular relevance as the authors suggest the recombinant SCGB3A was a source of contaminant endotoxin. In addition, the genetic background of the *Scgb3a2* KO mice was not listed. Were littermate controls used in in vivo studies? These important issues must be addressed.

6) Although LLC may not express high levels of IL-1β, they may express IL-18 which would be released if the NLRP3 inflammasome is engaged downstream of caspase-4/5/11. Can the authors confirm that the effect of SCGB3A2 in vivo is related to the pyroptosis of LLC cells, and not their production of IL-1β or IL-18?

7) Even in an LPS/endotoxin free condition, *Scgb3a2* KO mice exhibited profound LLC cell survival (Figure 1F, G, H, I). This clearly indicates that SCGB3A2 suppresses LLC cell proliferations beyond LPS-binding. This raises the question of whether SCGB3A requires LPS for its function (Figure 6F). If the authors mean to insist that microbiota source-derived LPS contributes to the phenomenon, a germ-free condition would have to be employed as a control. This issue needs to be addressed. In addition, to convince readers of anti-lung cancer function of SCGB3A and caspase-11, a more clinically relevant model such as *LSL-K-rasG12D* mice should be examined. Does deficiency of *Scgb3a2* or *Casp11* genes protect *K-ras* mutant mice? This could be immensely important for the relevance of the proposed mechanism.

---

## [Author Response]

Essential revisions:1) The main claim, that this pathway triggers pyroptosis, remains unproven at the genetic level. Although the authors indicate that LLC cells undergo pyroptosis following treatment with LPS and SCGB3A2 (Figure 5), the induction of cell death is low (Figure 5B and C). In addition, there is no evidence provided that supports the conclusion that death is caspase-11-dependent cell death (Figure 6F); this is a critical issue. Are Casp11 KO or Gsdmd KO LLC gRNA cells indeed resistant to LPS-induced cell death? Genetic deletion or depletion of caspase-4/5 or caspase-11 needs to be performed to confirm that the effect observed is pyroptosis due to LPS triggering the non-canonical inflammasome. The authors should also make an effort to assess whether inhibiting caspase-11 activation prevent pyroptosis and cell death. Finally, the authors should show that inhibition of clathrin mediated endocytosis (by Dynasore) inhibits killing of LLC. These experiments would add much-needed support to the overall premise outlined by the authors. As a side note, more standard assays for pyroptosis (such as LDH release) should be done.

We carried out several experiments as follows: (1) We established sh-casp-11-LLC cells and performed mouse xenograft model experiments using the same scheme as described in Figure 1. These in vivo results are now shown in Figure 4N. SCGB3A2 administration reduced the number of lung tumors when control LLC cells were used while the tumor numbers remained the same with or without SCGB3A2 administration when sh-casp-11-LLC cells were used. (2) We examined the effect of the inhibition of casp-11 activation using chemical compound “Wedelolactone”(Kobori et al., 2004) on LDH activity. LDH activity itself was significantly reduced by inhibition of *Casp11* signaling in LLC cells while no effect of LPS＋SCGB3A2 on LDH activity was observed (Figure 4M). (3) As suggested by the reviewer, we used Dynasore to inhibit clathrin-mediated endocytosis, and confirmed the importance of clathrin for LPS/SCGB3A2 endocytosis (Figure 4K), Casp-11 activation (Figure 4L), and LDH cytotoxicity activity (Figure 4M).

2) Does LPS given on its own suppress tumor growth in Scgb3a2(-/-) mice? How much of the tumor suppressive effect in Figure 1 is due to the LPS in the recombinant protein? For example, if SCGB3A2 suppresses tumors in mice how do SDC1 low tumors behave? Does knocking down or CRISPRing out SDC1 from LLC rescue them from LPS toxicity in vivo? The authors should CRISPR out TLR4 and SDC1 from LLC cells engraft them into their mouse model measure tumor load with and without LPS+SMGB3A2.

We calculated the endotoxin level in each recombinant SCGB3A2 preparation (see Supplementary file 1) and newly performed in vivo xenograft experiments using the amount of LPS contained in the protein preparation. The results, now shown in Figure 2F, clearly indicates that SCGB3A2 has much more important function than LPS alone. We have already demonstrated that sh-SDC1-LLC cells have reduced effects as compared with control cells when treated with SCGB3A2+LPS in vitro and with SCGB3A2 in vivo as shown in Figure 6A-D.

We made several attempts to establish sh-TLR4/SDC1-LLC cells. However, we were not able to obtain cells with both genes knocked down at the same time, at least in our system.

3) The authors concluded that NLRP3 mediates pyroptosis downstream of caspase-11 (Figure 6F). However, this contradicts the field's consensus that NLRP3 is dispensable for pyroptosis in non-canonical inflammasome signaling (Sharma and Kanneganti, 2016). This study does not address this contradiction. The authors need to address this discrepancy or they may wish to simply remove discussion of NLRP3, as it is highly unlikely that NLRP3 is involved in their system.

While it is clear that NLRP3 is upregulated by SCGB3A2+LPS treatment, we did not directly address a potential interaction between Casp-11 and NLRP3 in this manuscript. We decided to remove the discussion on NLRP3, and changed our model for the SGB3A2-LPS-SDC1 pathway as shown in Figure 6G in the revision.

4) The binding of LPS-SCGB3A2 was examined by agarose gel separation/reverse staining and DLS assay; the shift of LPS aggregate to lower sizes was shown (Figure 2B, C, and D). Given these findings, the authors concluded that LPS binds to SCGB3A. However, these shifts do not directly support the binding of LPS to SCGB3A2. For example, they may imply that SCGB3A has some LPS dissolution activity. The LPS-SCGB3A2 binding study should be re-examined by standard assays such as Surface Plasmon Resonance analysis and/or biotin-LPS pulldown. As a side note, the presented DLS assay is missing a SCGB3A alone control (Figure 2D).

We conducted pull down assay using biotin-LPS and streptavidin agarose gel. The result is now shown in Figure 2D in the revision.

We also added SCGB3A2 alone to the DLS assay graph in the revision (Figure 2E).

5) In the Materials and methods section, the procedure for recombinant SCGB3A protein preparation was not disclosed. This is of particular relevance as the authors suggest the recombinant SCGB3A was a source of contaminant endotoxin. In addition, the genetic background of the Scgb3a2 KO mice was not listed. Were littermate controls used in in vivo studies? These important issues must be addressed.

The purification method of recombinant human SCGB3A2 that we mainly used in this manuscript and that we will keep using for future studies has already been described (Cai et al., 2014). The information was added in the Materials and methods, Key resources table.

As for *Scgb3a2(-/-)* mice, they were backcrossed 10 times to C57BL/6N and the litter mates were used as controls. This is now stated in Materials and methods subsection “LLC cells mouse metastasis model”, and added in the Key resources table.

6) Although LLC may not express high levels of IL-1β, they may express IL-18 which would be released if the NLRP3 inflammasome is engaged downstream of caspase-4/5/11. Can the authors confirm that the effect of SCGB3A2 in vivo is related to the pyroptosis of LLC cells, and not their production of IL-1β or IL-18?

We performed IL-18 and IL-1β ELISA assays twice. In both cases, their levels were too low to be detected. Moreover, we showed that mRNA levels for IL18 in LLC cells was lower than that of B16F10 cells (Figure 6E). We believe that these results suggest that IL-18 is not critical for the LLC cells growth inhibition by SCGB3A2 in vivo.

7) Even in an LPS/endotoxin free condition, Scgb3a2 KO mice exhibited profound LLC cell survival (Figure 1F, G, H, I). This clearly indicates that SCGB3A2 suppresses LLC cell proliferations beyond LPS-binding. This raises the question of whether SCGB3A requires LPS for its function (Figure 6F). If the authors mean to insist that microbiota source-derived LPS contributes to the phenomenon, a germ-free condition would have to be employed as a control. This issue needs to be addressed. In addition, to convince readers of anti-lung cancer function of SCGB3A and caspase-11, a more clinically relevant model such as LSL-K-rasG12D mice should be examined. Does deficiency of Scgb3a2 or Casp11 genes protect K-ras mutant mice? This could be immensely important for the relevance of the proposed mechanism.

Unfortunately, we do not have access to a germ-free facility. We carried out in vivo metastasis model experiments using the amount of LPS that is found in various recombinant SCGB3A2 preparations as described in Supplementary file 1 without co-administration of SCGB3A2. We show in the revised Figure 2F and G that LPS alone, regardless of the amount used, did not show much effect in terms of lung tumor numbers as compared with mice administered SCGB3A2. More importantly, our in vitro results suggest that SCGB3A2 is an LPS binding protein and SCGB3A2+LPS upregulates Casp-11/NLRP3 expression. These results indicate that SCGB3A2 causes non-canonical inflammasome activation of LLC cells.

We also carried out lung carcinogenesis study using *CCSP-Cre;LSL-KrasG12D* mice that specifically express mutated *Kras* gene in lung. These mice were crossed with *Scgb3a2(fl/fl)* mice to produce *Ccsp-Cre;LSL-K-rasG12D;Scgb3a2(fl/fl)* andlittermate *Ccsp-Cre;LSL-K-rasG12D;Scgb3(fl/+)* mice. All these mice developed lung cancer by 4 months old. The survival of *Ccsp-Cre;LSL-K-rasG12D;Scgb3a2(fl/fl)* mice was significantly shorter than *Ccsp-Cre;LSL-K-rasG12D;Scgb3(fl/+)* mice. The results again indicate that SCGB3A2 is important to kill cancer cells. The new results were added to Figure 6F. Accordingly, the Results, and Materials and methods sections are modified. Also, Mitsuhiro Yoneda who carried out the lung carcinogenesis study was added as a co-author in the revised manuscript.

References

Sharma D., and Kanneganti, T.D. (2016) The cell biology of inflammasomes: Mechanisms of inflammasome activation and regulation J Cell Biol 213(6):617-29